# Hypoxia induced responses are reflected in the stromal proteome of breast cancer

Silje Kjølle [1], Kenneth Finne [1], Even Birkeland[1], Vandana Ardawatia [1], Ingeborg Winge[1], Sura Aziz [1,2], Gøril Knutsvik[1,2], Elisabeth Wik [1,2], Joao A. Paulo [3], Heidrun Vethe [1], Dimitrios Kleftogiannis [1,4] & Lars A. Akslen [1,2] ✉

Cancers are often associated with hypoxia and metabolic reprogramming, resulting in enhanced tumor progression. Here, we aim to study breast cancer hypoxia responses, focusing on secreted proteins from low-grade (luminal-like) and high-grade (basal-like) cell lines before and after hypoxia. We examine the overlap between proteomics data from secretome analysis and laser microdissected human breast cancer stroma, and we identify a 33-protein stromal-based hypoxia profile (33P) capturing differences between luminal-like and basal-like tumors. The 33P signature is associated with metabolic differences and other adaptations following hypoxia. We observe that mRNA values for 33P predict patient survival independently of molecular subtypes and basic prognostic factors, also among low-grade luminal-like tumors. We find a significant prognostic interaction between 33P and radiation therapy.

Reduced oxygen availability is a tumor microenvironment (TME) condition promoting cancer progress[1]. Hypoxia-inducible factor 1-alpha (HIF-1α) accumulates and leads to a range of adaptive processes, such as metabolic changes, tumor plasticity, immune evasion, angiogenesis, and metastasis[2]. Multiple target genes for HIF-1α have been reported, although cells may respond to hypoxia not exclusively through HIFs[3]. The complexity of hypoxia responses in human cancer tissues is not well studied at the proteomic level.

Intra-tumoral hypoxic regions often emerge as tumors outgrow their vascular supply, and hypoxia can trigger mechanisms like metabolic reprogramming and angiogenesis in the TME[4–6]. As an example, tumor vascular proliferation is linked to more aggressive subgroups of breast cancer[7]. Thus, hypoxia might represent a master regulator of several programs involved in tumor progression.

We investigate hypoxia responses in the breast cancer TME by combining cell secretomes (in vitro) with the tumor stromal proteome (in vivo), with particular attention to differences between luminal-like and basal-like tumor subtypes. This integrated approach of secretome and stromal analysis reveals distinct proteomic patterns.

## Results

To study the hypoxia response in breast cancer, we first analyzed the secretomes of four selected breast cancer cell lines (BCCL) derived from the two phenotypes (luminal-like, basal-like) by mass spectrometry-based proteomics (Fig. 1a). In the secretomes, we identified a total of 1787 secreted proteins, across all cell lines and oxygen conditions (Supplementary Fig. 1; see Supplementary Information and Supplementary Table 1 for details on selected cell lines).

### Secretomes are different between breast cancer subtypes at normoxia

We compared the luminal-like and basal-like secretomes at normoxia in the discovery BCCL panel ($n = 4$). The distribution and number of secreted proteins were similar for cells at normoxia and hypoxia (1211 and 1245 proteins, respectively) (Fig. 2a, b). At baseline, 331 proteins showed significantly higher levels from basal-like cell lines compared with luminal-like cells. Conversely, 133 proteins had significantly higher abundance in the luminal-like secretome.

By gene set enrichment analysis (GSEA), processes associated with more aggressive cancer, including metabolic changes, angiogenesis,

[1]Centre for Cancer Biomarkers CCBIO, Department of Clinical Medicine, Section for Pathology, University of Bergen, Bergen N-5021, Norway. [2]Department of Pathology, Haukeland University Hospital, Bergen N-5021, Norway. [3]Department of Cell Biology, Harvard Medical School, Boston, MA, USA. [4]Department of Informatics, Computational Biology Unit, University of Bergen, Bergen, Norway. ✉e-mail: lars.akslen@uib.no

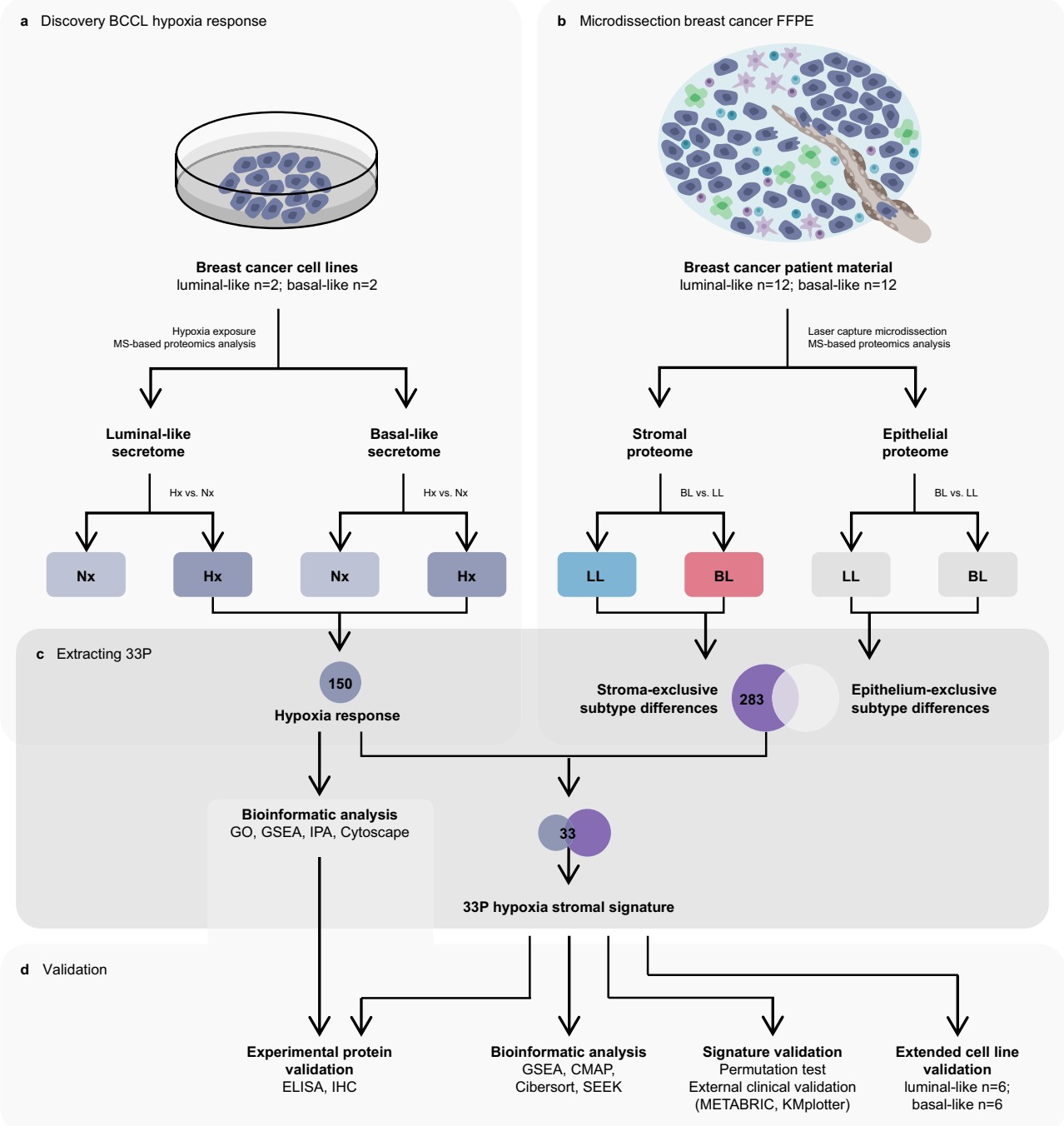

**Fig. 1 | Schematic overview of study with laboratory and data analysis.** Methods workflow for proteomics experiments of breast cancer cell line (BCCL) conditioned media (**a**) and formalin-fixed paraffin-embedded (FFPE) tumor samples (**b**). From hypoxic BCCL secretome experiments, 150 proteins showed hypoxia-increased secretion (Hx). From microdissected FFPE material, 283 proteins showed subtype differences only in the stromal compartment (basal-like (BL) vs. luminal-like (LL) subtypes). The 33-protein hypoxia stromal signature (33P) was generated from the overlapping proteins between the 150 hypoxia-increased proteins (BCCL secretome experiments) and the 283 proteins showing stroma-exclusive subtype differences (microdissected breast cancer patient material) (**c**). The hypoxia response proteins and 33P signature were validated using bioinformatic analysis and experimental validation (**d**). The hypoxia response proteins were investigated with bioinformatic analyses (gene ontology analysis (GO), gene set enrichment analysis (GSEA), ingenuity pathway analysis (IPA), network biology analysis using Cytoscape). The 33P signature was explored bioinformatically (GSEA, connectivity map analysis (CMAP), Cibersort, Search-Based Exploration of Expression Compendium (SEEK)) and validated with external clinical validation (METABRIC-Discovery, n = 852; KMplotter merged cohorts) for survival analysis and permutation test, and with extended cell line validation (BCCLs; LL n = 6, BL n = 6). 33P: 33-protein hypoxia stromal signature. BCCL breast cancer cell line. BL basal-like breast cancer subtype. CMAP Connectivity map analysis. ELISA enzyme-linked immunosorbent assay. FFPE formalin-fixed paraffin-embedded tissue. GO gene ontology analysis. GSEA gene set enrichment analysis. Hx hypoxia. IHC immunohistochemistry. LL luminal-like breast cancer subtype. MS mass spectrometry. Nx normoxia. SEEK search-based exploration of expression compendium.

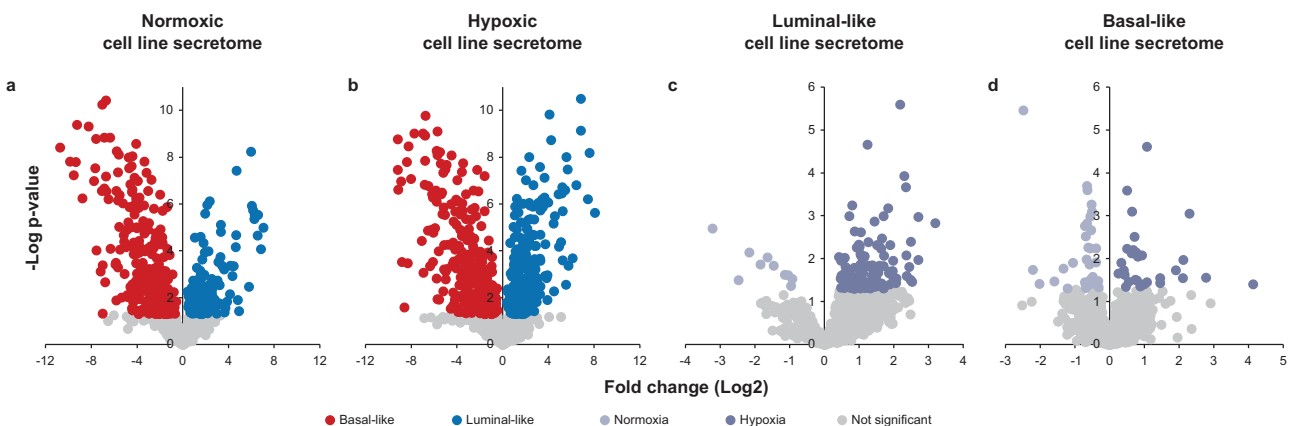

**Fig. 2 | Distribution of secreted proteins between oxygen conditions and subtypes in breast cancer cell secretomes.** Relative distribution of secreted proteins between the luminal-like ($n = 2$) and basal-like cell lines ($n = 2$) under normoxic (**a**) and hypoxic (**b**) conditions, and between the normoxic and hypoxic conditions from luminal-like (**c**) and basal-like cell lines (**d**). Colored circles represent significantly differentially secreted proteins between the comparisons (Two-sided Student's $t$ test, $p < 0.05$). Source data are provided as a Source Data file.

inflammation, immune responses, tissue remodeling and development, and cellular proliferation, were significantly enriched in the basal-like secretome compared with the luminal-like subtype (all FDR < 0.05) (Supplementary Data 1). This is in line with previously described differences at baseline between luminal-like and basal-like subtypes, based on mRNA and selected individual proteins[8-10].

Additionally, three of the PAM50 proteins (EGFR, CDH3, SLC39A6) were in common with the proteins separating basal-like and luminal-like secretomes at baseline. Of relevance for secretome studies, we found that 40 of the PAM50 signature genes/proteins have been reported in serum or plasma (plasma proteome database; PPD: http://www.plasmaproteomedatabase.org/)[11,12] and/or the Human Protein Atlas−blood protein (Human Protein Atlas proteinatlas.org)[13].

### Distinct hypoxia responses in breast cancer cells

We focused on secreted proteins being increased by hypoxia (hypoxome) (Fig. 2c, d); overall 150 proteins were significantly increased as compared to normoxia: 128 in luminal-like and 29 in basal-like cells (Supplementary Data 2). Only 7 proteins overlapped and showed increased abundance in both subtypes after hypoxia: CTSB, GAPDH, HNRNPF, RCN1, RNPEP, SDCBP, and VEGFA. The low number of hypoxia-upregulated proteins in common between the two breast cancer subtypes indicates that luminal-like and basal-like cells have distinct hypoxia responses.

Comparing the proteins separating basal-like and luminal-like secretomes under hypoxia, only one protein was overlapping with the PAM50 gene set (EGFR), indicating that the PAM50 classifier may be lacking hypoxic information for subtype stratification.

As we observed several intracellular proteins in our secretomes, we investigated cell viability and found this to be high, with no significant difference between cells conditioned at hypoxia and normoxia (average viability at hypoxia: 92.2%; normoxia: 93.9%), in either the luminal-like (hypoxia: 95.9%; normoxia: 96.5%; p=ns) or basal-like cell lines (hypoxia: 88.5%; normoxia 91.4%; p = ns; Mann−Whitney $U$ test). Further, gene ontology analysis of our secretome proteins showed significant enrichment of proteins in the extracellular region compared to random (GO:0005576; all proteins, FDR = 1.16 × 10^{−229}).

We then searched for key upstream transcription factors of the combined hypoxia response (luminal-like and basal-like) by the Ingenuity Pathway Analysis (IPA) program. Notably, HIF1A is associated with acute hypoxia response, whereas HIF2A is stabilized in chronic hypoxia[14]. Among the top five transcriptional regulators associated with the 150 hypoxia-induced proteins, we found MYC, TP53, ARNT, HNF4A, and HIF1A (ranked by strength of association) (Supplementary

Data 3). We found 15 of the 150 hypoxia-increased proteins to be HIF1A targets.

Next, using the IPA database combined with literature mining, we found that of the 150 hypoxia-increased proteins, Putative phospholipase B-like 2 (PLBD2) have not been previously associated with cancer. Based on sequence similarity, PLBD2 is a putative phospholipase, and probably involved in fatty acid metabolism. Studies are needed to elucidate the role of PLBD2 in cancer. Moreover, 40 of the 150 proteins have not been previously associated with breast cancer.

When IPA was performed separately for luminal-like and basal-like hypoxia responses (upregulated proteins), we found that MYC, TP53, HNF4A and ARNT were the top-ranked upstream transcriptional regulators for the luminal-like response, whereas TP53, ARNT and HIF1A were top-ranked for the basal-like response. Among the top five transcriptional regulators for each subtype, NRF2, encoded by the *NFE2L2* gene, was only found in the luminal-like hypoxic secretome, whereas TFEB and BCL6B were exclusively found for the basal-like response (Supplementary Data 3). Our findings indicate differences in luminal-like and basal-like hypoxia responses, and that these responses are not exclusively regulated by hypoxia-inducible factors (HIFs).

Next, we investigated a STRING-generated[15] interaction network for the 150 hypoxia-increased proteins, and found higher number of interactions and/or associations compared to a random reference set (PPI enrichment $p$-value $< 1.0 × 10^{−16}$) (Fig. 3; Supplementary Data 2); 125 of the 150 proteins were associated to at least one other protein in a large main network; the 125 proteins showed overrepresentation of proteins involved in metabolic processes (GOID 8152, $p = 8.9 × 10^{−18}$) and included angiogenesis (e.g., VEGFA, ANG, ANGPTL4; GOID 1525, $p = 6.6 × 10^{−4}$) (Supplementary Data 4). The network showed subclusters associated with metabolic processes such as glycolysis (e.g., GAPDH, LDHA, MDH2; GOID 6096, $p = 1.2 × 10^{−4}$) and TCA cycle (e.g., IDH2, MDH1, ACO1; GOID 6099, $p = 5.8 × 10^{−9}$) (Supplementary Table 2).

### Hypoxia-secretomes are enriched in proteins associated with energy metabolism

We explored differences in the hypoxia response (upregulated proteins) within luminal-like and basal-like subtypes separately. The luminal-like hypoxome was mainly enriched in processes related to metabolism, such as glycolysis (21 of 62 gene set proteins, $p = 0.02$), TCA cycle (12 of 32 gene set proteins, $p = 0.04$), and oxidative phosphorylation (9 of 35 gene set proteins, $p = 0.05$) (Fig. 4a−f; Supplementary Data 5). Lactate dehydrogenase (LDHA), a key enzyme in anaerobic glycolysis, was significantly increased in the

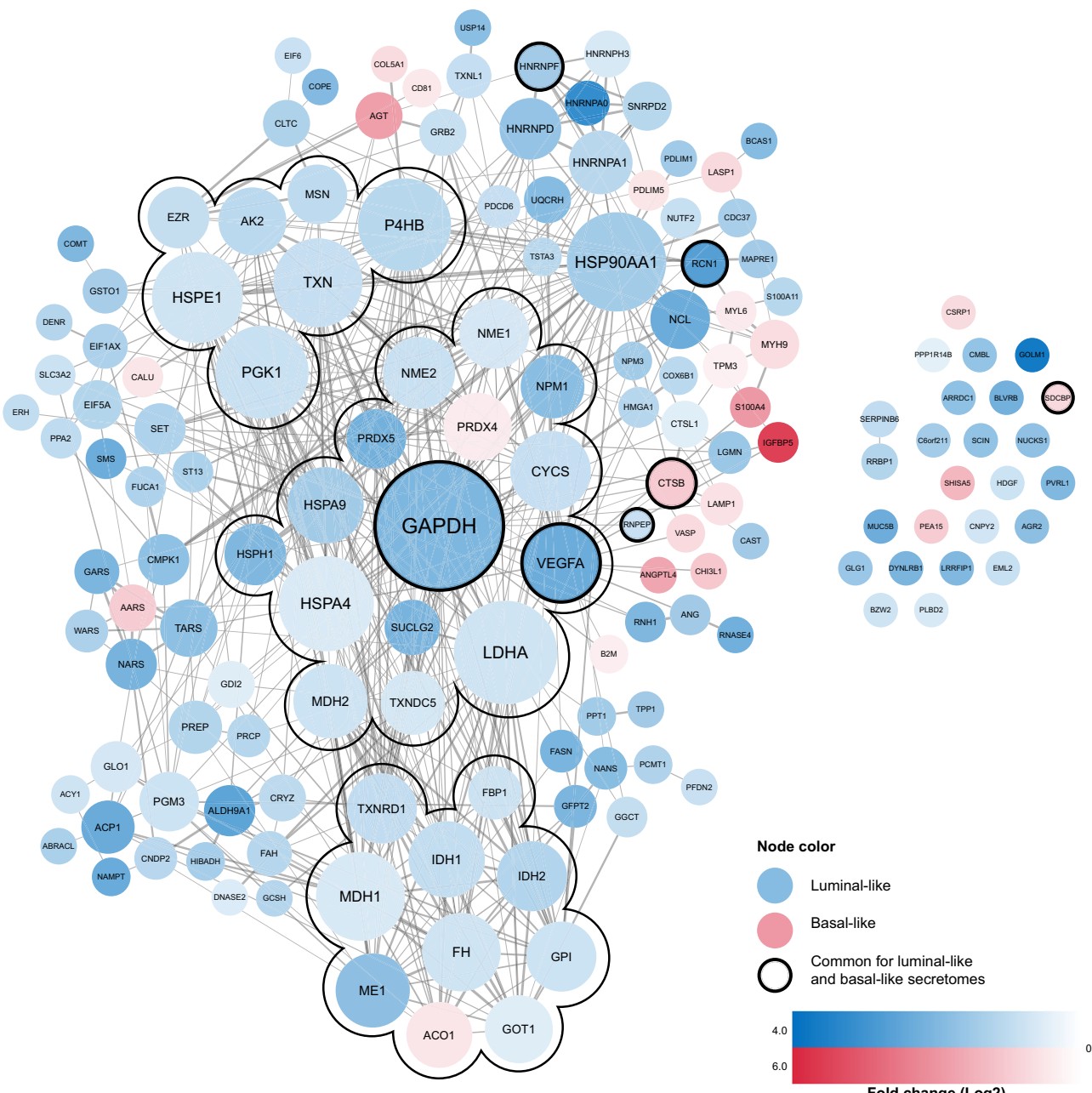

**Fig. 3 | Interaction network of hypoxia response proteins.** Protein-protein association network of the 150 hypoxia-upregulated proteins (PPI enrichment $p$-value < $1.0 \times 10^{-16}$). Blue nodes represent luminal-like hypoxia response proteins, and red nodes represent basal-like hypoxia response proteins. The color intensity of the nodes represents fold change (Log2) compared to normoxic conditions, within each subtype (luminal-like cell lines, $n = 2$; basal-like cell lines, $n = 2$). The nodes with black borders represent proteins with increased secretion in response to hypoxia in both subtypes, in which the

color represents the subtype with largest fold change. Node size represents the number of connections (degree), with larger node sizes representing a higher number of connections. The network contains three subclusters of highly interconnected proteins (MCODE 1.4.2) with overrepresentation of TCA cycle, glycolysis, and cell redox homeostasis, consisting of 10, 15 and 7 proteins, with 36, 52 and 12 connections, respectively, connecting the nodes within each subcluster.

hypoxic secretome of luminal-like cells, but not in basal-like cell lines (Supplementary Data 2). Whereas we did not observe significant hypoxia-induced differences in energy metabolism among basal-like cells, these cells still showed 1.9-fold higher levels of LDHA at normoxia compared to the luminal-like hypoxic secretome ($p = 0.002$). In contrast, the basal-like hypoxome showed enrichment related to tissue development, immune responses, inflammation and secretion (Supplementary Data 6). Our findings suggest that luminal-like cells have a stronger hypoxia response, while basal-like cells may have adapted to a hypoxic environment in vivo, as hypoxic and necrotic

regions are more frequent in rapidly growing tumors, such as basal-like breast cancers.

## Hypoxia-secretomes are enriched in proteins associated with angiogenesis

Hypoxia is associated with expression and/or secretion of angiogenic proteins being targets of HIFs[4]. We compared angiogenic proteins between hypoxic and normoxic conditions and observed that the global basal-like secretome (normoxic and hypoxic) included 52 angiogenesis-related proteins (GOID 1525), while the luminal-like

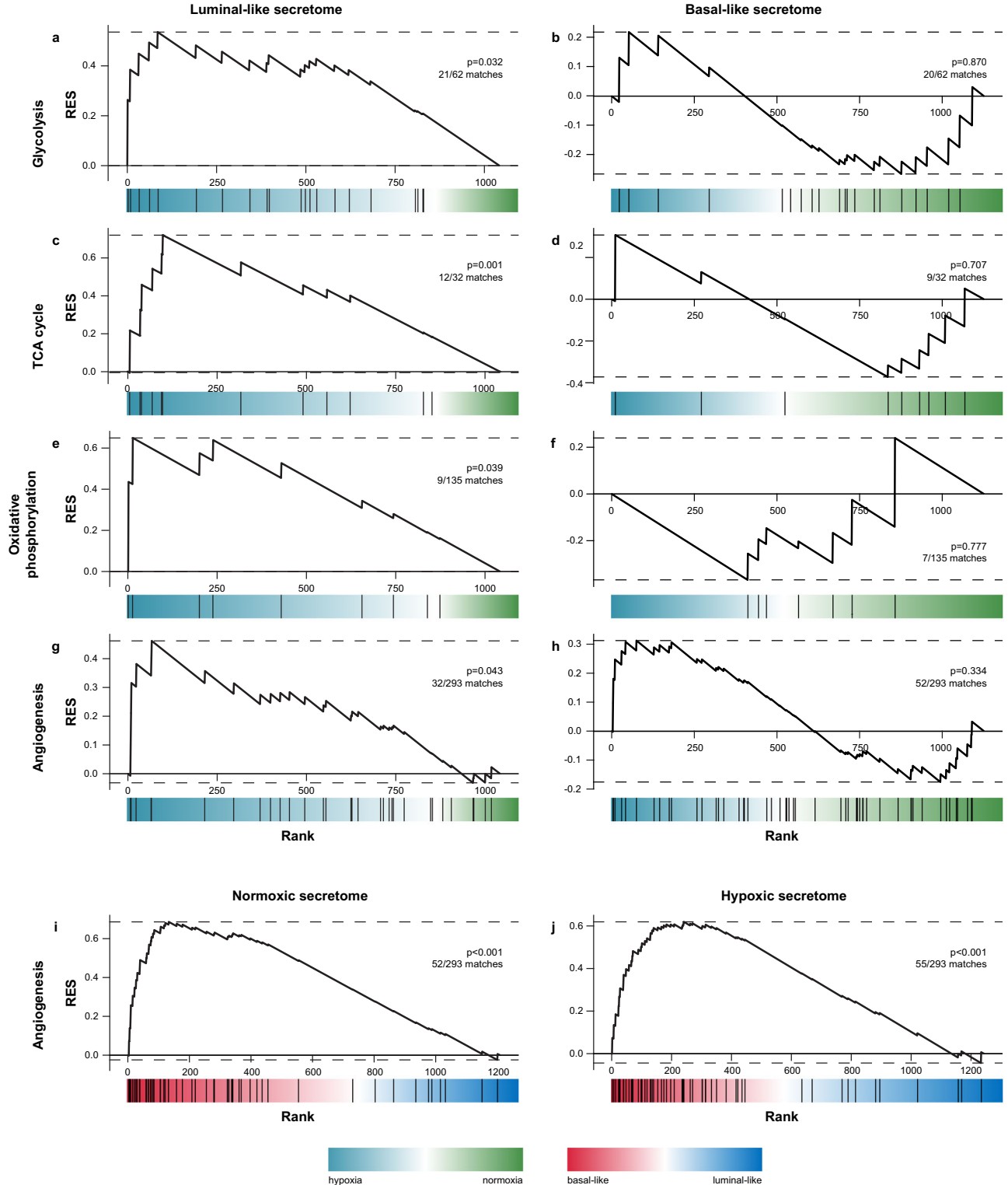

**Fig. 4 | Comparing hypoxic and normoxic secretomes in luminal-like and basal-like breast cancer cell lines.** Gene set enrichment analysis of secretome data ranked using a two-sided t-test from hypoxia-increased (blue) to hypoxia-decreased (green) in luminal-like (**a**, **c**, **e**, **g**) and basal-like (**b**, **d**, **f**, **h**) cell lines. The selected analyses show significant enrichment of KEGG pathways glycolysis, TCA cycle, oxidative phosphorylation and angiogenesis (GOID 1525) in the luminal-like hypoxic secretome. The basal-like hypoxic secretome was not enriched in either of the gene sets. Ranking the secretome data from basal-like (red) to luminal-like (blue) under normoxic (**i**) and hypoxic (**j**) conditions, showed an enrichment in angiogenic proteins in the basal-like subtype in both oxygen conditions. *P*-values were not adjusted for multiple testing. GSEA gene set enrichment analysis. RES running enrichment score. KEGG Kyoto Encyclopedia of Genes and Genomes. GO gene ontology analysis.

**Table 1 | Angiogenesis-related proteins upregulated by hypoxia in breast cancer cell lines**

| Gene name | Protein name | Fold change after hypoxia | |
|---|---|---|---|
| | | Luminal-like | Basal-like |
| ANG | Angiogenin | 2.9[a] | 1.7 |
| ANGPTL4 | Angiopoietin-like 4 | 1.6 | 6.1[a] |
| MYH9 | Myosin-9 | 2.3 | 2.0[a] |
| NCL | Nucleolin | 5.5[a] | 0.8 |
| PDCD6 | Programmed cell death protein 6 | 2.0[a] | 2.1 |
| PRCP | Lysosomal pro-X carboxypeptidase | 2.4[a] | 1.1 |
| VEGFA | Vascular endothelial growth factor-A | 5.5[a] | 1.9[a] |
| WARS | Tryptophan--tRNA ligase | 2.6[a] | 1.7 |

[a]Significantly increased in response to hypoxia (two-sided Student's $t$ test, significance level 0.05).

secretome revealed 32 proteins involved in angiogenesis (Fig. 4g, h). Of the 32 luminal-like matches, 28 were in common with the 52 basal-like angiogenic proteins. The luminal-like, but not the basal-like secretome, showed significant hypoxia-induced enrichment of angiogenesis-related proteins (luminal-like: $p = 0.043$; basal-like: $p = 0.33$). However, several of the basal-like angiogenic proteins were already higher at baseline, compared with hypoxia-increased luminal-like angiogenic proteins, including ANG, NCL, PRCP and VEGFA (Supplementary Data 2).

We then compared angiogenic proteins in luminal-like and basal-like secretomes, both at normoxia and after hypoxia (Fig. 4i, j) and found enrichment of angiogenic proteins in the basal-like secretomes in both oxygen conditions (normoxia: $p < 0.001$; hypoxia: $p < 0.001$). These data indicate discrete angiogenic responses within luminal-like and basal-like cells following hypoxia.

Among the 150 hypoxia-increased proteins, only 8 were associated with angiogenesis (Table 1). Notably, vascular endothelial growth factor A (VEGFA) was the only angiogenesis-related protein that was increased by hypoxia in both luminal-like and basal-like cells. VEGFA showed 3.7-fold higher abundance in normoxic basal-like secretomes compared with hypoxic luminal-like secretomes ($p = 0.01$); this difference was validated by enzyme-linked immunosorbent assay (ELISA) (Supplementary Fig. 2) (see also Fig. 1d). Lysosomal Pro-X carboxypeptidase (PRCP), a key regulator of vascular homeostasis, was significantly increased in luminal-like secretomes only, but showed 7.4-fold higher secretion from normoxic basal-like cells compared with hypoxic luminal-like cells ($p < 0.001$). Angiopoietin-like 4 (ANGPTL4) and myosin-9 (MYH9) were the only two angiogenic proteins that were hypoxia-increased only in the basal-like cell lines (Table 1). Secretome levels of ANGPTL4 were evaluated by ELISA for validation and showed the same patterns as in MS data (Supplementary Fig. 2).

Further, cathepsin B (CTSB), being connected to angiogenesis[16], showed higher levels of secretion from baseline basal-like compared to hypoxic luminal-like cell lines, as well as being hypoxia-increased in both subtypes (Supplementary Data 2). This was validated by ELISA and showed similar patterns for different cell lines when examined separately. The luminal-like levels of CTSB were not detected by ELISA, being consistent with the lower secretion from luminal-like than basal-like cells (Supplementary Fig. 2).

Taken together, our data indicate differences in the secretion of angiogenic proteins following hypoxia between luminal-like and basal-like cells, with only VEGFA overlapping between subtypes. Luminal-like cells increase their secretion of angiogenesis-promoting factors after hypoxia to a greater extent compared with basal-like cells, although

several of the basal-like proteins were considerably higher at baseline. This might suggest that basal-like cancer cells are already in an activated angiogenic-like state at baseline.

## Integrated proteomics analysis of cell line secretomes and microdissected tumor stroma from human breast cancer

Our data indicate that luminal-like and basal-like cells have distinct hypoxia responses, and that basal-like cells may be characterized by more features related to hypoxia present at baseline. As secreted and released proteins from tumor cells are part of the TME in vivo and important in promoting aggressive TME characteristics, we predicted that such proteins might be identified in the stromal compartment of human breast tumors. We separated tumor stroma and tumor epithelium by laser capture microdissection of formalin-fixed paraffin-embedded luminal-like and basal-like breast cancer samples, followed by shotgun proteomics analysis of the extracted proteins (Fig. 1b). In the tumor cell compartment, 4157 proteins were detected, compared to 2150 proteins in stromal samples (Supplementary Fig. 1).

Among stromal proteins, the majority were also found in tumor epithelium. We then focused on proteins that were significantly different between luminal-like and basal-like tumor stroma. Proteins differing significantly between the luminal-like and basal-like subtypes in the tumor epithelium were then subtracted from the set of proteins that differed in the tumor stroma compartment. This resulted in 283 proteins that represented significant and unique differences between the subtypes in the stromal compartment; 202 proteins with significantly higher abundance in basal-like stroma; 81 proteins with significantly higher abundance in luminal-like stroma.

Of interest, six proteins (FOXA1, ERBB2, MAPT, NAT1, PHGDH, KRT5) and one protein (PHGDH) overlapped between PAM50 and the differentially expressed proteins between basal-like and luminal-like subtypes in microdissected tumor epithelium and stroma, respectively. This illustrates that the PAM50 signature is mainly tumor epithelial cell-based.

When exploring the 283 proteins by gene ontology analysis, we found a significant overrepresentation of proteins in the cellular components 'Extracellular matrix' (GOID 31012, FDR = $6.60 \times 10^{-15}$) and 'Extracellular space' (GOID 5615, FDR = $1.37 \times 10^{-57}$), as well as involvement in processes of 'Extracellular matrix organization' (GOID 30198, FDR = $5.03 \times 10^{-5}$) and 'Collagen fibril organization' (GOID 30199, FDR = $7.81 \times 10^{-4}$).

We cross-referenced the 283 proteins with the 150 hypoxia-upregulated proteins from our secretome studies, revealing 33 overlapping proteins that were differentially abundant in both datasets (Fig. 1c). This protein set was termed the 33P hypoxia stromal signature (33P) (Supplementary Table 3). As reflected by the 150 hypoxia-upregulated proteins, 33P was also overrepresented by proteins involved in glycolysis (GOID 6096, FDR = 0.0136), TCA cycle (GOID 6099, FDR = 0.0132), and other carbohydrate metabolic processes (FDR < 0.05).

To examine the uniqueness of this 33P signature compared to a random selection of 33 proteins from a pool of the 150 and 283 proteins (above), we performed a random selection permutation analysis and found that 33P was significantly stronger than expected by random chance ($p < 0.0001$) (Supplementary Fig. 3) (see also Fig. 1d).

Further, to illuminate potential associations between 33P and specific cell types in the TME, we used Cibersort[17] to deconvolute bulk transcriptomic data from METABRIC-Discovery ($n = 852$). We inferred the immune cell abundance for a subset of patients with basal-like and triple-negative breast cancer[18]. Basal-like tumors were stratified using the 33P signature score (Q1–Q3 vs. Q4), and we observed lower number of B-cells and CD8-cells in the worse outcome (Q4) subgroup of 33P, indicating potential immune suppression (Supplementary Fig. 4). Notably, we found fewer resting mast cells and an increase in activated mast cells associated with higher 33P. Our findings indicate an

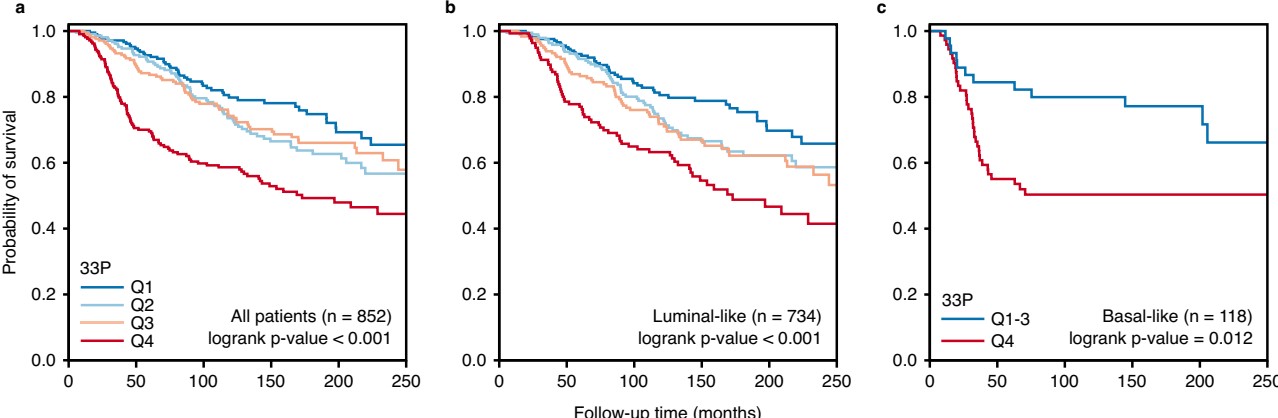

**Fig. 5 | Survival plots in breast cancer patients scored by 33P hypoxia stromal signature (METABRIC-Discovery cohort).** Kaplan–Meier of breast cancer specific survival in patients diagnosed with luminal-like and basal-like breast cancer ($n = 852$) (**a**), only luminal-like subtype ($n = 734$) (**b**), and only basal-like subtype ($n = 118$) (**c**) in the METABRIC-Discovery cohort. The patients are divided into quartiles depending on 33P signature score (33P-low, Q1 in blue; 33P-high, Q4 in red). The plots show a significant association between high 33P scores (Q4) and poor survival for patients diagnosed with luminal-like and basal-like breast cancer. Survival differences between groups were evaluated with a two-sided log-rank test.

association between 33P and immune cell levels within the basal-like subtype.

As one of the main strengths of secretome studies is the potential presence of such proteins in serum or plasma, we examined the 33P in the PPD[11,12] and found 32 of the 33 signature proteins (not in PPD: COPE). We further explored the Human Protein Atlas—blood protein[13] and found all signature proteins to be detected in plasma by MS analysis.

### High 33P mRNA score associates with aggressive breast cancer features

We explored whether features of tumor cell hypoxia reflected in the stroma, as indicated by 33P, were associated with aggressive breast cancer phenotypes and patient outcome. For this, we included 852 patients diagnosed with luminal A, luminal B or basal-like breast cancer in the METABRIC-Discovery cohort and extracted normalized expression values (mRNA) of genes corresponding to 33P proteins. High 33P mRNA score (by upper quartile) associated with large tumor size, high histologic grade, lymph node metastases, ER negative tumors, and a basal-like phenotype (Supplementary Fig. 5).

In a recently published proteomics cohort ($n = 209$)[19], the 33P score was significantly associated with molecular breast cancer subtypes (Supplementary Fig. 5). High 33P was associated with high histologic grade ($p < 0.001$; grade 3 vs. 1-2) and high tumor cell proliferation by Ki67 expression ($p < 0.001$).

In addition to basic prognostic factors, 33P correlated with independent signatures and gene sets for tissue hypoxia[20–22] (Supplementary Fig. 6); correlations were also significant after removing overlapping proteins (all $p < 0.001$) (Supplementary Table 4) with similar Spearman's rank correlation coefficients (Halle: $\rho = 0.35$, previously $\rho = 0.40$; Eustace: $\rho = 0.59$, previously $\rho = 0.64$). Strong correlations with other hypoxia signatures support that 33P reflects hypoxic features being present in the stromal compartment.

A high 33P mRNA score was associated with signatures for proliferation[23,24], glycolysis (Hallmark glycolysis, MSigDB), angiogenesis by vascular proliferation[25,26], epithelial-to-mesenchymal transition (EMT)[27], and stemness[28–30] (see below, and Supplementary Fig. 6).

We applied the search-based exploration of expression compendium (SEEK) and found that 33P associated with triple-negative phenotype ($p = 0.0006$) and high-grade breast cancer ($p < 0.00001$) in two datasets (GSE45255.GPL96 and GSE4922.GPL96), as well as p53 mutations (GSE22093.GPL96; $p = 0.038$); the p53 association was also found in METABRIC-Discovery, including among luminal A cases ($p = 0.02$).

33P was higher in tumor tissue compared with normal tissue (GSE15852.GPL96) ($p = 0.001$).

### High 33P mRNA score associates with reduced patient survival

High 33P was associated with decreased breast cancer-specific survival (log-rank test, $p$-value $< 0.001$) (Fig. 5a), also when stratifying by molecular subtype (Fig. 5b, c). Notably, stratification of the luminal-like category showed that high 33P was associated with reduced survival within the luminal A category (log-rank test, $p$-value $= 0.02$). Conversely, basal-like tumors were significantly stratified by 33P, with clearly better survival for those with lower values (Q1-3). Notably, high 33P was associated with shorter survival in the merged cohorts from KMplotter, and when stratified by subtypes (Supplementary Fig. 7).

By multivariate survival analysis, 33P demonstrated independent prognostic value when adjusting for molecular subtype (luminal-like or basal-like; by PAM50), as well as the basic prognostic factors tumor diameter, histologic grade and lymph node status (Cox' regression, Wald test, $p = 0.001$) (Table 2), and also when stratifying the cohort by molecular subtype (Supplementary Table 5).

### Relation between 33P and treatment

To explore the potential interaction between 33P and various treatments, we applied the retrospective observational METABRIC-Discovery cohort ($n = 852$) with information on endocrine treatment, chemotherapy, and radiation therapy. We initially performed stratified survival analyses (with/without treatment), and we found no difference for endocrine treatment or chemotherapy with respect to 33P, while different survival patterns were present for radiation therapy (yes vs. no) (Fig. 6a, b). For those treated with radiotherapy, low 33P (lower quartile) was associated with significantly better survival than high values (upper quartile). Statistically, we found a significant interaction with radiotherapy for the prognostic value of 33P ($p = 0.02$; HR = 1.93 [1.21–3.30]), also after adjustment for basic factors (tumor diameter, histologic grade, lymph node status). The diverging effect of radiotherapy was significant also in patients with luminal A breast cancer (Fig. 6c, d). Whether 33P might be applied to stratify patients for radiotherapy, would need to be studied in a prospective randomized clinical trial for verification. Of note, patients having radiotherapy had significantly higher histologic grade, and were more often ER negative and lymph node positive.

We then asked whether any of the 33 proteins were more important than others in terms of their impact on patient survival. Using the METABRIC-Discovery dataset ($n = 852$), we applied a

reduction algorithm, assuming that not all proteins in 33P would be equally strong. The 33P signature was reduced by recursively leaving one gene/protein out and then testing the predictive strength of the remaining N-1 genes/proteins in a survival analysis (Q1-3 vs. Q4). The strongest N-1 signature (lowest log-rank p-value) was retained, and the process was repeated until only one gene remained. The reduced version of 33P with the strongest effect on survival ($p = 4.3 \times 10^{-17}$, compared to baseline 33P $p = 1.0 \times 10^{-8}$) was these 18 proteins: CDC37, COL5A1, CTSB, GAPDH, GRB2, HNRNPA1, HNRNPD, HNRNPF, HSPA4, HSPA9, IDH1, LDHA, MYL6, P4HB, PGK1, RRBP1, SET, VASP (Supplementary Fig. 8). These 18 proteins showed a strong separation of the upper quartile patients (Q4) in the luminal A subgroup, and this

prognostic impact was validated in KMplotter ($p < 1.0 \times 10^{-16}$; $n = 2032$), also in the luminal A subgroup ($p = 0.00015$; $n = 631$).

## Signatures reflecting metabolic processes, vascular proliferation and cellular plasticity are increased in 33P-high breast cancer

To investigate and validate the ability of the 33P signature to reflect metabolic reprogramming of the TME, GSEA was performed on the METABRIC-Discovery cohort, with proteins ranked from 33P-high to 33P-low. Gene sets reflecting glycolysis and other metabolic processes were significantly enriched in 33P-high tumors (all FDR < 0.05). Glycolysis was overrepresented among 33P proteins (GOID6096, $p = 0.0009$), and a gene set reflecting glycolysis was top ranked and significantly enriched in 33P-high tumors by GSEA (rank 2, FDR < 0.0002, Hallmark glycolysis, MSigDB) and significantly correlated with 33P in the METABRIC-Discovery cohort (Supplementary Fig. 6).

A gene set reflecting VEGF signaling was significantly enriched by GSEA in 33P-high tumors (MSigDB, C6 oncogenic signature VEGF_A_UP.V1_DN, FDR < 0.0001), and validated by independent signatures reflecting VEGF and vascular proliferation (Supplementary Fig. 6). Further, 33P significantly correlated with signatures reflecting epithelial-mesenchymal transition and stemness features, including a signature for high Nestin expression[27–29]. Notably, 33P correlated positively with a mammary stem cell score and a luminal progenitor signature, and negatively with a mature luminal signature[30] (Supplementary Fig. 6). Our findings suggest that hypoxia is related to more stem-like features.

We expanded the characterization of 33P by performing a STRING-analysis (string-db.org)[15], and found very strong connectivity between the proteins; 29 of 33 proteins (88%) were included in one large network (Supplementary Fig. 9). The 150 hypoxia proteins that 33P is derived from also showed high connectivity (83% in one large network) (Fig. 3). Notably, we found that 9 of the 33P proteins were associated with the 'VEGFA-VEGFR2 signaling pathway' (WikiPathways, $p < 0.001$).

Regarding angiogenesis, we have validated our findings using an in-house breast cancer tissue cohort and found that 33P (by MS-proteomics) was positively associated with vascular proliferation by IHC, a marker of activated angiogenesis[7] ($n = 42$; $p = 0.05$).

## Gene expression propose compounds with potential relevance to 33P-high breast cancer

To search for biologically relevant targets in 33P-high breast cancer, we queried the drug signature database Connectivity Map (CMAP version 02)[31] for compound-related gene expression profiles negatively enriched in 33P-high tumors, as such compounds may contribute to decrease some of the features associated with high 33P scores. Among

**Table 2 | Multivariate survival analysis (proportional hazards regression model) of breast cancer patients (METABRIC-Discovery cohort)**

| Variable | n | Univariate analysis | | Multivariate analysis | |
|---|---|---|---|---|---|
| | | HR (95 % CI) | p-value | HR (95 % CI) | p-value |
| *Tumor size* | | | | | |
| < 20 mm | 263 | 1.00 | <0.0005 | 1.00 | 0.006 |
| > 20 mm | 589 | 2.05 (1.47–2.86) | | 1.62 (1.15–2.28) | |
| *Histologic grade* | | | | | |
| 1-2 | 443 | 1.00 | <0.0005 | 1.00 | 0.070 |
| 3 | 409 | 1.86 (1.41–2.46) | | 1.33 (0.98–1.80) | |
| *Lymph node status* | | | | | |
| Negative | 453 | 1.00 | <0.0005 | 1.00 | <0.0005 |
| Positive | 399 | 2.25 (1.70–2.98) | | 1.83 (1.38–2.45) | |
| *PAM50 subtype* [a] | | | | | |
| Luminal-like[b] | 734 | 1.00 | <0.0005 | 1.00 | NS |
| Basal-like | 118 | 1.91 (1.37–2.68) | | 1.26 (0.87–1.82) | |
| *33P hypoxia stromal signature* | | | | | |
| Q123 | 639 | 1.00 | <0.0005 | 1.00 | 0.001 |
| Q4 | 213 | 2.120 (1.60–2.81) | | 1.67 (1.23–2.27) | |

Statistical test: Two-sided Wald test. Adjustment for multiple testing was not performed.
*CI* confidence interval, *HR* hazard ratio, *n* number of patients, *NS* not significant.
[a]Only patients with luminal A, luminal B and basal-like breast cancers were included (n=852, METABRIC-Discovery cohort).
[b]Luminal-like: patients with luminal A and luminal B breast cancer subtypes.

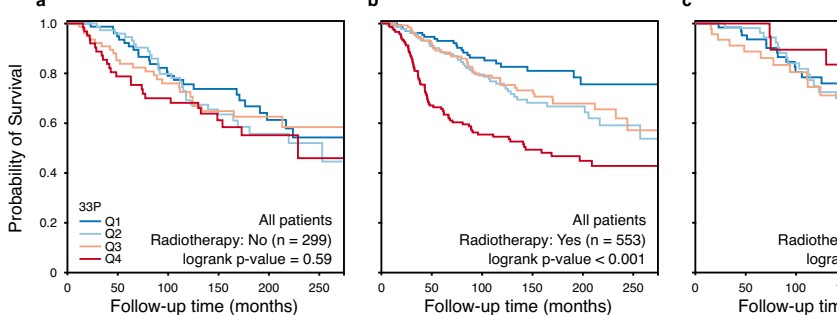
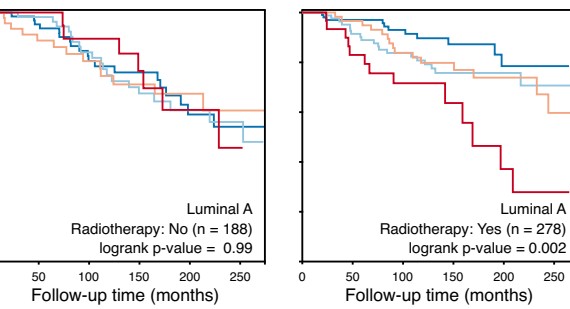

**Fig. 6 | Interaction between the 33P signature and radiotherapy (METABRIC-Discovery cohort).** Univariate survival analysis (Kaplan-Meier method) of breast cancer-specific survival in patients from the METABRIC-Discovery cohort. The patients were grouped into four 33P score quartiles (33P-low, Q1 in blue; 33P-high, Q4 in red). The left panels (**a**, **b**) include all patients (luminal-like and basal-like; $n = 852$); the right panels include only luminal A patients ($n = 466$) (**c**, **d**). The patients were stratified into patients that did not (**a**, **c**) or did (**b**, **d**) receive radiotherapy. Survival differences between groups were evaluated with a two-sided log-rank test.

1309 small molecules represented in CMAP, expression profiles from compounds with properties promoting attenuation of tumor effects from hypoxia were top ranked (Supplementary Data 7). Previous studies on many of these compounds have demonstrated anti-hypoxia effects in cancer (e.g., resveratrol[32], sirolimus[33]). Several of the top-ranked compounds have also been shown to have antioxidant effects and/or effects on the transcription factor NRF2 (nuclear factor erythroid 2-related factor 2), encoded by the *NFE2L2* gene (e.g., apigenin[34]). NRF2, found in our IPA analysis of upstream transcription factors for luminal-like hypoxia response proteins, is a known regulator of genes containing antioxidant response elements[35,36].

In stratified CMAP analyses (luminal-like and basal-like separately), gene expression profiles of compounds with PI3K/mTOR inhibitory properties were top-ranked and negatively enriched in 33P-high tumors (Supplementary Data 7). Adding to this, signatures reflecting PI3K/AKT/mTOR activation were top-ranked and significantly enriched in tumors with high 33P (mRNA) score (GSEA/MSigDB; H and C6 subsets; FDR < 0.05). Taken together, results from CMAP analyses, used as a hypothesis-generating/supporting tool, propose a biological relevance of NRF2 activating and/or PI3K/mTOR inhibitory compounds to 33P-high tumors.

### Immunohistochemical expression of NRF2 in tumor tissue
Based on results from IPA and CMAP analyses, IHC was performed to examine NRF2 expression in the tumor stromal and epithelial compartments using a breast cancer cohort of 42 cases with tissue proteomics information and 33P status. Stromal NRF2 expression (Supplementary Fig. 10) was found to be significantly correlated to the 33P signature scores ($\rho = 0.56$, $p < 0.001$), supporting our IPA findings (above); epithelial NRF2 expression was not associated with 33P.

### Validation of 33P by expanded cell line experiments
To validate 33P derived from the original 4 cell lines (2 luminal-like, 2 basal-like), we added 8 additional cell lines (4 luminal-like, 4 basal-like) in a new validation experiment that included all 12 cell lines (Supplementary Fig. 11; Fig. 1d). We predicted that this expansion would implicate a wider biologic spectrum with increased diversity and better coverage of hypoxia responses. First, we performed a discovery analysis on the validation experiment (12 cell lines), similar to the initial discovery, which resulted in a set of 36 proteins (36P) that correlated significantly with 33P (correlation coefficient 0.70; $p < 0.001$; Pearson) (Supplementary Fig. 12); the correlation was significant also when overlapping proteins ($n = 10$) were omitted from 36P (correlation coefficient 0.50; $p < 0.001$; Pearson).

Next, we investigated the expression of the 33P proteins in our new dataset (12 cell lines) and found (by GSEA) that 33P was significantly associated with hypoxia (Supplementary Fig. 13). When examining individual proteins from 33P in the validation dataset, 13 of these proteins were significantly altered in either the luminal-like or basal-like cell lines (Supplementary Table 3). A subscore consisting of these 13 proteins (13P) was generated in the METABRIC-Discovery cohort (mRNA data), showing significant association with tumor subtypes and patient survival among all luminal-like or basal-like breast cancers ($n = 852$), as well as in the luminal A category ($n = 466$) (Supplementary Fig. 14). This result is similar to what was observed in the initial proteomic discovery data (resulting in 33P), indicating that 13P represents a consistent subset of hypoxia-altered proteins across the entire and expanded cell line panel ($n = 12$). Notably, 13P was slightly stronger than 33P when directly compared in a multivariate analysis of patient survival (METABRIC-Discovery) (Supplementary Data 8), in particular among luminal A cases, indicating that 13P is capturing a broader range of hypoxia responses and might reflect a wider set of aggressive stromal characteristics. The 13P score (Q4, upper quartile) was also associated with reduced survival in the KMplotter dataset (Supplementary Fig. 14), and 13P was significantly associated with

response to radiation therapy ($p = 0.035$, test for interaction). Our data support the validity of the original 33P signature. At the same time, the 13P signature, based on the extended experiments, showed slightly stronger prognostic impact.

## Discussion
Hypoxia is a master driver of tumor progression[1,2], and extensive work has been performed on hypoxia-inducible factors (HIFs) and their target genes. However, less is known about the hypoxia responses at the global proteomic level in cancer, and whether these are relevant in a clinical context. Although hypoxia has been investigated using MS-based proteomics, the main focus has been on cellular proteins and pathways[37]. Even with an increasing number of cancer secretome studies, also including hypoxic secretomes[38,39], many of these have concentrated on extracellular vesicles and their role in metastasis[40,41].

Here, we explored secretome proteins from breast cancer cells and response patterns following hypoxia, with particular attention to differences between luminal-like and basal-like subtypes. We found almost completely discrete hypoxia responses between these subgroups. Whereas luminal-like cells showed a marked response with increased secretion of proteins related to metabolic changes and angiogenesis activation, basal-like cells were less affected, possibly because of higher levels of hypoxia-associated proteins present at normoxia. Our data suggest that basal-like cells might have a chronic hypoxia-like phenotype compared to luminal-like cells.

In the luminal-like secretome, metabolic processes in particular, and angiogenesis, were enriched after hypoxia. The luminal-like hypoxome showed enrichment of proteins related to glycolysis, the TCA cycle, and oxidative phosphorylation, compared to the normoxic secretome. Also, LDHA was significantly elevated. Our findings support an increased reliance of glycolysis in luminal-like breast cancer cells in adapting to hypoxic conditions[42] and reinforce the importance of the Warburg effect[43,44]. In the basal-like hypoxome, proteins involved in tissue development, immune responses and inflammation were enriched. In line with this, basal-like cells are prone to cellular plasticity, as also seen during tissue and organ development, with increased cell migration and immune evasion[45,46]. These cells are more likely to disseminate when exposed to hypoxia[30], parallel to the more aggressive disease behavior observed among patients with such tumors[47]. Whether a baseline hypoxic phenotype among basal-like cells can be reversed, is not clear[48,49].

Hypoxia induces increased secretion of angiogenic factors from tumor cells[50,51], and angiogenesis is higher in basal-like than in luminal-like tumors[8,52]. Here, luminal-like cells showed a significant enrichment of secreted angiogenesis-related proteins under hypoxic conditions, whereas most such factors already showed higher levels in the basal-like secretome at normoxia.

In our cell line experiments, we observed several of the secretome proteins that are normally found in the intracellular compartment. Although this could potentially reflect cell death, we found high viability with no significant difference between hypoxic and normoxic conditions when stratified by luminal-like and basal-like subtypes. Notably, intracellular cytosolic proteins may be contained in extracellular vesicles, and intracellular proteins may have extracellular functions. Also, secretion modes may be different in cancer cells compared to non-cancerous cells, and this could possibly explain why we detect intracellular proteins in our secretome data[53].

By using an integrative design combining global proteomics information from hypoxic cell line secretomes in vitro with micro-dissected human breast cancer stroma, to interrogate hypoxic responses in the TME in vivo, we identified a 33-protein signature (33P) that indicated more high-grade stromal features within subgroups of breast cancer. The 33P score was associated with reduced patient survival beyond that of tumor cell-based PAM50 breast cancer classification. Even among luminal A tumors, 33P defined a subgroup of

more aggressive cancers, indicating that hypoxia and "high-grade stroma" can evolve in some luminal-like tumors which are most often low-grade by other classifiers.

We asked whether 33P was related to treatment modalities. Based on the retrospective and observational METABRIC-Discovery cohort, 33P showed a significant interaction with radiation, and prognostic separation was observed among patients receiving radiotherapy. Apparently, cases with low levels of 33P had improved prognosis following radiation treatment. This differential effect of radiotherapy was also significant in patients with luminal A breast cancer. Whether 33P can be applied to stratify breast cancer patients for radiotherapy, should be further studied in randomized clinical trials for verification. Our data is in line with studies demonstrating reduced effect of radiation therapy in hypoxic cancers[54,55].

Based on our extended cell line experiment, we found that a subset of the 33 proteins, 13P, was also prognostically significant and slightly stronger compared with 33P, in particular among luminal A cases. This might indicate that 13P reflects a somewhat broader set of aggressive stromal features associated with hypoxia. Notably, 13P showed a similar interaction with radiation therapy. This shorter signature may be more easily translated to a clinical assay, for example by using mRNA values from whole tissue specimens, similar to our validation studies.

NRF2, considered a master regulator of cellular antioxidant response[35], was identified as an upstream regulator of luminal-like hypoxia response (by IPA analysis), and stromal NRF2 protein expression by IHC was associated with high 33P in our breast cancer cohort. Using the CMAP database for drug response and repurposing exploration, NRF2 (by activation) and PI3K (by inhibition) were pointed out as potential targets in hypoxic tumors[56]. From our IPA analysis, we would expect inhibition of upstream transcription factors of our hypoxia-increased proteins, including NRF2, and it is not clear how to explain our results. Regarding the PI3K/mTOR pathway, also indicated by our CMAP exploration, only few clinical trials have tested PI3K and/ or mTOR inhibitors in advanced triple negative breast cancer, with published data from one trial so far (i.e., NCT00499603), and with no change in response rate (at 12 weeks) following downregulation of mTOR[57]. Whether stratification by 33P would have provided more information, is not known.

There are some limitations to the present study. Although our findings indicate that hypoxia responses reflected in the stroma associate with aggressive tumor features, the involved mechanisms must be explored in more detail, including the role of NRF2. This also relates to targets for anti-hypoxia treatment, and whether hypoxia-associated features could be reversed or blocked. Regarding protein distribution, many aspects would need to be clarified, like the role of vesicular transport and true secretion in cancer cells, in addition to other mechanisms. Finally, our finding that 33P relates to radiation therapy should be followed up in clinical trials for verification.

Taken together, our findings at the proteomic level suggest that hypoxia profiling in human breast cancer reveals distinct responses of metabolic and other adaptive changes, and that differences are present between tumor subgroups. Importantly, 33P should be further explored in relation to radiotherapy.

## Methods
### Selection of breast cancer cell lines
Selection of breast cancer cell lines (BCCL) for the discovery phase (4 BCCLs; luminal-like $n = 2$, basal-like $n = 2$) and the extended validation experiments (8 additional BCCLs; luminal-like $n = 4$, basal-like $n = 4$) was based on literature studies and bioinformatic mapping[58,59]. By combining mapping of existing literature information with in-house bioinformatics analyses (below), we provide stronger evidence on the molecular suitability of candidate cell lines, for the selection of cell lines and for extended validation experiments. This information is

summarized in Supplementary Table 1. The initially selected luminal-like cell lines are both ER and PR positive, and both selected basal-like cell lines are triple-negative. These cell lines were selected with a balance between primary and metastatic source (Supplementary Table 1). The selected cell lines are widely used and included in several large studies investigating breast cancer cells in vitro[60-65]. All selected cell lines are part of at least one of American Type Culture Collection (ATCC)'s cell line panels for breast cancer or triple-negative breast cancers, and none of the included cell lines are among the cell lines with debated subtype or characteristics (e.g., SKBR3, previously classified as luminal[60,62], and later classified as HER2-enriched[61]).

To identify representative cell lines for the validation panel, we performed an unbiased exploratory analysis using publicly available transcriptomic ($n = 54$) and proteomic ($n = 28$) data from the Cancer Cell Line Encyclopedia (CCLE)[58,59]. For both transcriptomic and proteomic datasets, we used the available gene expression and protein expression matrices as input. The cell lines were projected into the 2D space using multidimensional scaling (MDS) (Supplementary Fig. 11).

The cell lines formed clusters, and the clusters were strongly driven by their molecular subtype identity. This information was used as a guide to assess differences in the expression profiles of the available cell lines ($n = 4$), and unbiasedly select new candidate cell lines to cover the observed 2D space (validation cell lines, $n = 8$). We believe that the original four cell lines were neither outliers nor expressing very different transcriptomic or proteomic profiles from all other cell lines. Instead, they were quite representative in the 2D subtype space, as were the additional 8 cell lines that we subsequently selected.

Expanding the cell line panel of luminal-like cell lines, we decided to include a HER2-positive cell line consistent with the luminal B tumor subtype, and three cell lines with hormone receptor status patterns corresponding to luminal A tumors. Importantly, regarding the HER2-positive cell lines included in our study (initial: BT-474; additional: ZR-75-30); these cell lines are hormone receptor positive and have luminal characteristics, and belong to the luminal category of cell lines.

Expanding the cell line panel of basal-like cell lines, three basal A cell lines and one basal B and claudin-low cell line were included to also have a balance between basal A and basal B cell lines in follow-up experiments. Importantly, all six basal-like cell lines were triple-negative. The basal A cell lines were included as this category is corresponding closely with the basal-like tumor subtype[9,66], and the basal B category of cell lines were included since these are more similar to the triple-negative tumors.

When selecting cell lines for the validation experiment, we carefully selected cell lines with similar media and supplements to ensure that there was no obvious external metabolic bias between the luminal and basal-like subtypes.

All cell lines were provided from American Type Culture Collection (ATCC) with certificate of analysis. All cell lines tested negative for mycoplasma contamination.

### Selection of patients and study approval
For the in-house human tumor samples used in our study (for microdissection and proteomics, $n = 24$; for immunohistochemistry, $n = 42$; see below), the protocol was approved by the Western Regional Committee for Medical and Health Research Ethics, REC West (REK #2014/1984). The informed consent was waived by the REC West Committee, based on national guidelines, as well as the age and size of the full cohort covered by the approval. However, the actual patients included were informed about the research project and the possibility to withdraw. All studies were performed in accordance with guidelines and regulations by the University of Bergen and REK, and in accordance with the Declaration of Helsinki Principles.

Tumor tissues ($n = 24$) were collected from female patients (aged 50–69 years) diagnosed with breast carcinoma NST (no special type)

during 1996–2003, as part of a prospective and population-based screening program. Sex was defined by the national and unique 11-digit personal identification number. Tissue sections from 24 primary tumors (12 basal-like, 6 luminal A, 6 luminal B) were included for microdissection; tumor categories were based on the St Gallen 2013 classification[67]. All basal-like samples were also triple-negative, and all luminal samples were estrogen and progesterone positive, and HER2-negative. The luminal B tumors displayed more than 15% Ki67-positive nuclei[68].

## Cell cultures

For the discovery experiments, BT-474 (ATCC® HTB-20™) was grown in RPMI medium, MCF7 (ATCC® HTB-22™) and Hs 578T (ATCC® HTB-126™) were grown in DMEM medium and MDA-MB-231 (ATCC® HTB-26™) cells were grown in F-12 medium. All cell lines were supplemented with 10% fetal bovine serum (FBS), 1% penicillin strep-tomycin (PS) and 1% L-glutamine. In addition, MDA-MB-231 were supplemented with 1% glucose. For the extended validation panel of BCCLs, the additional cell lines (HCC1428 (ATCC® CRL-2327™), T47D (ATCC® HTB-133™), ZR751 (ATCC® CRL-1500™), ZR-75-30 (ATCC® CRL-1504™), MDA-MB-468 (ATCC® HTB-132™), HCC1143 (ATCC® CRL-2321™), HCC1187 (ATCC® CRL-2322™), BT-549 (ATCC® HTB-122™)) were cultured according to recommended protocols from ATCC. The cell lines were maintained at 37 °C in a humidified atmo-sphere with 5% $CO_2$, and all work was performed in a sterile environ-ment. Cells were sub-cultured at approximately 80% confluency by washing with PBS and incubation with trypsin (0.25%) and dividing into new cell culture flasks with fresh medium. Number of cells and viability were calculated using Countess™ Automated Cell Counter (Invitrogen).

## Conditioned media

The cell lines were grown to approximately 80% confluency in 175 cm² flasks, washed with PBS three times, and covered with basic medium without additives. The cells were incubated in normal conditions for 1 h, before the washing procedure was repeated. Then, 15 mL basic medium was added (no additives) and the cells were incubated for 24 h at either normoxia (21% $O_2$, 5% $CO_2$) or hypoxia (1.2% $O_2$, 5% $CO_2$). After 24 h, the conditioned medium was transferred to tubes and cen-trifuged at 3000 $g$ for 5 min to remove cell debris, and the supernatant was stored at −80 °C.

## Enzyme-linked immunosorbent assay

ELISA was performed on conditioned media for validation of the MS data on vascular endothelial growth factor A (VEGF-A; Quantikine® ELISA Human VEGF Immunoassay, R&D Systems™, DVE00), angiopoietin-like 4 (ANGPTL4; DuoSet® ELISA Development system Human Angiopoietin-like 4, R&D Systems™, DY3485), and cathepsin B (CTSB; DuoSet® ELISA Development system Human Total Cathepsin B, R&D Systems™, DY2176). ELISA analysis was performed after the manufacturer's protocol, and results were normalized to total protein concentrations.

## Microdissection of human breast cancer samples

Ten micrometers thick formalin-fixed paraffin-embedded (FFPE) sec-tions were deparaffinized, rehydrated and stained with hematoxylin. Breast cancer epithelium and tumor stroma (adjacent non-epithelial tissue) were laser capture microdissected (PALM MicroBeam, Zeiss) and pressure catapulted into a tube cap (AdhesiveCap 500 opaque, Zeiss). Tumor epithelium and tumor stroma areas were selected under supervision of an experienced breast pathologist (L.A.A), using digital high-resolution images of parallel sections stained with hematoxylin-eosin. Depending on availability (0.5–1.9) × $10^7$ µm³ tissue was obtained.

Subsequently, to estimate the purity of microdissected samples, we compared the intensities of the epithelial marker cytokeratin-8 in the tumor epithelial and the tumor stromal samples after proteomics analysis (Supplementary Fig. 15). We found on average 62-fold higher intensities of cytokeratin-8 in the tumor epithelium compared to the tumor stroma fraction, respectively (basal-like: 68-fold, $p = 3.2e−7$; luminal-like: 56-fold, $p = 7.5e−12$). By estimation, on average, only 1.6% (median: 1.7%) epithelial tissues were present in the stromal samples. The low levels of epithelium in microdissected stroma were true for both basal-like and luminal-like samples; the luminal-like samples had on average 6.1-fold higher content of cytokeratin-8 compared to basal-like samples in tumor epithelium ($p = 3.9e−5$). This was as expected since cytokeratin-8 is higher in luminal compared with basal-like epithelial cells.

## Sample preparation and mass spectrometry analysis

Conditioned media samples were concentrated using 3 kDa Amicon® Ultra-15 Centrifugal Filter Units (Merck, Kenilworth, NJ, USA) and lyo-philized using a vacuum concentrator. The protein pellet was dis-solved in 8 M urea/20 mM methylamine solution and protein concentration was estimated using Qubit™ Protein Assay Kit (Thermo Fisher Scientific). For secretome samples, 10 µg protein from each sample was prepared and total volume was adjusted. Reduction of proteins was performed by adding 4 µL of 100 mM dithiotreitol (DTT), incubating 1 h, room temperature (RT). Followed by alkylation by adding 5 µL of 200 mM iodoacetamide (IAA), incubating for 1 h, RT, in dark. Proteins were digested using a 1:50 ratio of trypsin to protein concentration and incubated overnight at 37 °C. The trypsin reaction was stopped by adding 15 µL of 10% formic acid (FA) to each sample. The microdissected patient tissue were prepared with the filter-aided sample preparation (FASP) protocol[69]. In short, the microdissected patient tissue was lysed in 4% SDS, 100 mM DTT and 100 mM Tris/HCl pH8. The lysate was then centrifuged to remove cellular debris and the protein sample was loaded onto a Microcon 30 kDa centrifugal filter unit (Merck Millipore, MA, USA). The samples were washed (8 M Urea, 0.1 M Tris PH 8.5), alkylated (0.1 M IAA) and washed again, first with urea, then three times with 50 mM ammonium bicarbonate. Finally, the proteins were digested on the filter unit using trypsin in a ratio 1:50 trypsin:protein. The resulting peptides were collected by centrifuga-tion. After digestion, all samples were desalted using Oasis HLB µElu-tion plates (Waters, Milford, MA, USA), and lyophilized. Prior to mass spectrometry analysis, conditioned medium samples for discovery experiments were dissolved in 0.1% FA solution, patient samples in 2% acetonitrile (ACN)/0.1% FA solution, and conditioned medium samples for extended validation experiments were dissolved in 5% ACN/5% FA. The peptide concentration of the conditioned media samples was estimated using NanoDrop™.

## LC-MS/MS analysis

Conditioned media samples from the discovery BCCL panel were analyzed during a 60 min gradient on an LTQ-Orbitrap Elite mass spectrometer (Thermo Fisher Scientific, Waltham, MA, USA) coupled to a Dionex Ultimate 3000 RSLC system. The peptides were separated on a 15 cm × 75 µm analytical column (Acclaim PepMap 100 ID nano-Viper column) packed with 2 µm C18 beads. The microdissected sam-ples were analyzed in their entirety during a 180 min gradient on a Q-Exative HF mass spectrometry (Thermo Fisher Scientific), coupled to a Dionex Ultimate NCR-3500 RSLC system. The peptides were sepa-rated on a 25 cm × 75 µm analytical column (PepMap RSLC, EASY-spray column) packed with 2 µm C18 beads). The MS was operated in data-dependent acquisition (DDA) mode. Raw data were acquired through the Xcalibur software (Thermo Fisher Scientific).

Mass spectrometry data for conditioned media samples from the validation BCCL panel were collected using the Exploris 480 mass spectrometer (Thermo Fisher Scientific, San Jose, CA) coupled with a Proxeon 1200 Liquid Chromatograph (Thermo Fisher Scientific). Peptides were separated on a 100 µm inner diameter microcapillary

column packed with ~25 cm of Accucore C18 resin (2.6 μm, 150 Å, Thermo Fisher Scientific). We loaded ~1 μg onto the column.

Peptides were separated using a 90 min gradient of 3 to 25% acetonitrile in 0.125% formic acid with a flow rate of 520 nL/min. The scan sequence began with an Orbitrap MS[1] spectrum with the following parameters: resolution 120,000, scan range 350–1350 Th, automatic gain control (AGC) target "standard", maximum injection time "auto", RF lens setting 40%, and centroid spectrum data type. We selected the top twenty precursors for MS[2] analysis which consisted of HCD high-energy collision dissociation with the following parameters: resolution 15,000, AGC was set at "standard", maximum injection time "auto", isolation window 1.2 Th, normalized collision energy (NCE) 28, and centroid spectrum data type. In addition, unassigned and singly charged species were excluded from MS[2] analysis and dynamic exclusion was set to 90 s.

## Computational analysis of proteomics data

Raw MS files from discovery BCCL panel secretomes and micro-dissected tissues were analyzed using MaxQuant[70] (v1.5.3.30 for conditioned medium samples and v1.6.0.16 for patient samples) with label-free quantification and "match between runs" enabled. The precursor ion tolerance for total protein level profiling was set to 20 pmm, and product ion tolerance to 0.5 Da. Carbamidomethylation of cysteines was set as fixed modifications, and oxidation of methionines and N-terminal acetylation was set as variable modifications. The false discovery rate (FDR) for peptide and protein identification was set to 1%. MS/MS spectra were searched in the Andromeda search engine against the forward and reverse Human UniProt database.

The validation BCCL panel secretomes, raw data were processed using the FragPipe (v18) proteomics pipeline software, wherein peptide identification was performed with MSFragger (v3.5)[71] with precursor and fragment mass tolerance in peak matching was set to 20 ppm. Peptide validation was performed with Percolator (v3.05)[72], and protein inference was done by ProteinProphet from the Philosopher toolkit (v4.4.0)[73]. MS1 quantification was performed using IonQuant (v1.8)[74] with the "Match between runs" option enabled. MaxLFQ protein intensity algorithm was selected, and intensities were normalized between experiments. Mass-to-charge (*m/z*) ratio tolerance were set to 10 ppm.

The identified proteins were analyzed using Perseus[75] (v1.6.0.2 for the discovery BCCL panel secretomes and microdissected tissue, and v2.0.7.0 for validation panel BCCL secretomes); the data were grouped into luminal-like or basal-like, and in addition hypoxia or normoxia for conditioned medium samples. Proteins with valid quantification in less than 50% of samples in at least one group were removed for analysis of the discovery panel of BCCL and patient samples. Non-filtered data from the extended BCCL panel was used for validation. Imputation was used to replace missing values (from a normal distribution: width 0.3, downshift 1.8) for secretome samples in the discovery panel. A two-sample Student's *t* test was performed to compare the groups, and a p-value significance threshold was set to 0.05.

Gene ontology analyses were performed using Panther Classification System[76] (PANTHER14.0, Overrepresentation Test, GO Ontology database released 2019-01-01). Gene sets significantly enriched in the hypoxic secretome were explored by applying the Gene Set Enrichment Analysis (GSEA; www.broadinstitute.org/gsea)[77] and signatures from Molecular Signatures Database (MSigDB; www.broadinstitute.org/gsea/msigdb), using the fgsea (version 1.15.0) R-package[78].

Protein network analyses were performed using StringDB[15], Cytoscape[79] (v3.5.1), and the Cytoscape add-on MCODE[80] (v1.4.2). Subcluster analysis was done in MCODE with the following settings: network scoring: include loops: false, degree cutoff: 2; cluster finding: node score cutoff: 0.2, haircut: true, fluff: false, K-Core: 2, max. depth from seed: 100.

The upstream regulator analysis was generated by QIAGEN's Ingenuity Pathway Analysis program (IPA®, QIAGEN Redwood City, www.qiagen.com/ingenuity). Settings for IPA were as follows: Expression Analysis with ´Exp Log Ratio´ values, Reference set (Ingenuity Knowledge Base, Genes Only), Confidence (Experimentally Observed), and for Node Type, Data Source, Species, Tissue & Cell Lines and Mutations, we selected all.

## Signature discovery

The signature proteins were derived from integrated analysis of secretomes from discovery BCCL and microdissected stromal tissue proteomics data. The proteins that were in common for the hypoxia-increased proteins (hypoxia vs. normoxia) and the stroma-exclusive subtype differences (basal-like vs. luminal-like) were extracted as the protein signature (see Fig. 1c). The signature proteins were validated in the extended validation panel of BCCLs.

## Signature scoring

Each signature gene was normalized by subtraction, i.e., the average gene expression value (all patient samples) was subtracted from the expression value of each patient sample. The signature score was calculated by summing the normalized expression values for each signature gene.

## Gene expression analysis of patient cohorts

For the exploration of gene expression patterns related to the 33P signature score in breast cancer, the signature was mapped to publicly available mRNA datasets with additional information on clinico-pathologic and follow-up data and molecular tumor subtypes, defined by the PAM50 algorithm[81] (METABRIC-Discovery cohort[82], *n* = 852; HER2 and normal-like subtypes were excluded). The online database "KM plotter" (www.kmplot.com)[83] was also applied to evaluate the 33P mRNA score in relation to recurrence-free breast cancer survival in a merged dataset of 3951 (updated *n* = 4934) breast cancer cases. The cut-off point for analyses (upper quartile) with dichotomized 33P mRNA score values was defined after considering frequency distributions and survival pattern of quartiles.

Gene sets significantly enriched in cases with high 33P mRNA score were explored by applying the Gene Set Enrichment Analysis (GSEA; www.broadinstitute.org/gsea)[77] and signatures from Molecular Signatures Database (MSigDB; www.broadinstitute.org/gsea/msigdb), using J-Express (version 2012)[84]. Multiple probes covering the same gene were collated according to max probe[77]. Genes differentially expressed between tumors of high versus low 33P mRNA score were identified based on Significance Analysis of Microarrays[85].

For comparisons, we analyzed separate signature scores reflecting effects of hypoxia[20–22], scores reflecting proliferation[23,24], glycolysis (MSigDB, HALLMARK_GLYCOLYSIS), angiogenesis by vascular proliferation[25,26], epithelial-mesenchymal transition (EMT)[27], signatures reflecting stemness features[28,29], and luminal progenitor and mature luminal signature scores[30].

## Connectivity Map analysis of drug signatures

We explored correlations between the global gene expression pattern of breast cancers with high 33P mRNA score and drug signatures in the Connectivity Map (CMAP) database[86] (METABRIC-Discovery cohort). As a basis for the CMAP analyses, we included genes differentially expressed (FDR < 0.006; fold change ≥1.5 or ≤−1.5) between tumor subsets of low and high 33P mRNA scores (cut-off point upper quartile).

## Proteomic analysis of external patients

We downloaded the recently published proteomic dataset on breast cancer by Asleh and colleagues[19] to explore associations between 33P and clinico-pathologic features (luminal-like and basal-like cancers

only; n = 209)[19]. The 33P signature was scored as described above. The heatmap was generated using the ComplexHeatmap R-package (v2.15.1)[87]. Boxplots were generated using ggplots2.

## Immunohistochemical staining

Immunohistochemistry detection of NFR2 expression in tissue samples was performed manually on 4–5 μm thick tissue microarray (TMA) sections from formalin-fixed paraffin-embedded tumor tissue from an in-house cohort of breast cancer patients (n = 42; luminal-like 23, basal-like 19) with MS-proteomics information in parallel. Briefly, target retrieval for NRF2 was performed in Ventana Benchmark Ultra staining platform (Roche Tissue Diagnostics, Ventana Medical Systems, USA) with Cell Conditioning (CC1, #06414575001, Roche Tissue Diagnostics, Ventana Medical Systems, USA) (pH9) at 95 °C for 48 min before endogenous peroxidases were blocked with Inhibitor CM (from DAB-kit #5266645001, Roche Tissue Diagnostics, Ventana Medical systems) at 37 °C 4 min. Slides were incubated with a monoclonal rabbit antibody against NRF2 (Clone EP1808Y, ab62352, Abcam, USA, diluted 1:100) for 60 min, followed by incubation with EnVision rabbit HRP (#K400311-2, Agilent, USA) for 30 min. To add color at the site of target antigen recognized by the primary antibody, DAB chromogen (#K346811-2, Agilent, USA) was applied for 10 min. Finally, sections were rinsed in distilled water and counterstained with Haematoxilin (#S330130-2, Agilent, USA).

NRF2 staining was recorded using a semi-quantitative and subjective grading system, considering the intensity of staining (none = 0, weak = 1, moderate = 2, and strong = 3) in tumor stromal and epithelial areas separately[88]. The NRF2 antibody was validated by the manufacturer in both positive and negative cells (HELA) and tissue samples (human pancreatic carcinoma and human kidney cancer tissue) with known localization patterns to confirm specificity and sensitivity, and in-house breast cancer and placenta tissues were established as positive controls.

## Statistical analyses of patient data

Data were analyzed using SPSS (Statistical Package of Social Sciences), Version 25.0 (Armonk, NY, USA; IBMM, Corp). A two-sided p-value less than 0.05 was considered statistically significant. A p-value of 0.05–0.10 was considered to be of borderline statistical significance (trend). Categories were compared using Pearson's chi-square or Fisher's exact tests when appropriate. Non-parametric correlations of bivariate continuous variables were tested by Spearman's rank correlation test. Spearman's rank correlation coefficient (ρ) is reported. Mann–Whitney U and Kruskal–Wallis tests were used for comparing continuous variables between groups. Odds ratios (OR) and their 95% confidence intervals were calculated by the Mantel–Haenszel method.

For survival analyses, the endpoint was death from breast cancer. Follow-up time was defined as the time from the date of diagnosis to the date of death or last follow-up. Univariate survival analysis by the Kaplan-Meier method was performed using the log-rank test. Patients who died of other causes or who were alive at last date of follow-up were censored. The influence of co-variates on breast cancer-specific survival was analyzed by Cox' proportional hazards multivariate method and tested by the Enter method. All variables were tested by log-minus-log plots to determine their ability to be incorporated in multivariate modeling. When categorizing continuous variables, cut-off points were based on median or quartile values, also considering the distribution profile, the size of subgroups, and number of events in survival analyses.

## Cibersort analysis

CIBERSORT[17] is a tool that uses gene expression data to estimate the cell type abundances in a mixed cell population. In our study, we used deconvoluted immune cell type abundances from METABRIC cohort performed by/generated by Craven and colleagues[18].

## SEEK analysis

Search-Based Exploration of Expression Compendium (SEEK)[89] is a search engine for transcriptomic data, providing thousands of expression datasets from published studies. SEEK implements a computational method that takes as an input a set of queries, genes, and returns a robust ranking of co-expressed genes whilst it ranks and prioritizes relevant expression datasets. In our study, we used SEEK with the 33P signature as input. We explored the top-ranked datasets, and we used the available information for extra validations of our findings.

## Reporting summary

Further information on research design is available in the Nature Portfolio Reporting Summary linked to this article.

## Data availability

The mass spectrometry proteomics data generated in this study have been deposited to the ProteomeXchange Consortium[90] via the PRIDE partner repository[91] (http://www.ebi.ac.uk/pride). The secretome data for the discovery panel of BCCLs are available via ProteomeXchange with the dataset identifier PXD027136. The microdissected patient material data are available via ProteomeXchange with identifier PXD027012. The secretome data for validation panel of BCCLs are available via ProteomeXchange with identifier PXD040532. Mass spectrometry data were searched against the forward and reverse Human UniProt database (https://www.uniprot.org/proteomes/UP000005640; downloaded/accessed 2016-01-08 (discovery BCCL panel), 2022-11-21 (validation BCCL panel), 2017-10-22 (microdissected patient material)). Clinical data on patients used for tissue micro-dissection might be made available for researchers on a request that does not include revelation of identifiable patient information, upon completion of a Data Transfer Agreement and confirmation of ethical approval. This study included analysis of data from the publicly available METABRIC-Discovery cohort[82] (available from the European Genome-Phenome Archive, Dataset ID: EGAD00010000210; unique identifier: https://doi.org/10.1038/nature10983), and the proteomic dataset from Asleh et al. Nature Communications (2022)[19] available from the supplementary information (unique identifier: https://doi.org/10.1038/s41467-022-28524-0). Survival analysis for hypoxia signatures was performed using the online KMplotter analysis platform (https://kmplot.com/analysis/)[83]. Publicly available data from the Cancer Cell Line Encyclopedia (CCLE) was used in this study. Processed transcriptomic data from breast cancer cell lines are available from CCLE[58] and they are accessible via the depmap portal (https://depmap.org/portal/download/all/). CCLE proteomic data[59] are available from https://gygi.hms.harvard.edu/publications/ccle.html. CIBERSORT analysis data from Craven et al.[18] are available from https://github.com/kelgalla/tnbctils or https://doi.org/10.5281/zenodo.4542590. The SEEK[89] database (https://seek.princeton.edu/seek/) was used for search-based exploration of the identified proteins (publicly available datasets: GSE45255.GPL96[92], unique identifier: https://doi.org/10.1186/gb-2013-14-4-r34; GSE4922.GPL96[93], unique identifier: https://doi.org/10.1158/0008-5472.CAN-05-4414; GSE22093.GPL96[94,95], unique identifiers: https://doi.org/10.1093/jnci/djq524, https://doi.org/10.1371/journal.pone.0049529; GSE15852.GPL96[96], unique identifier: https://doi.org/10.1016/j.prp.2009.11.006). The remaining data are available within the article, supplementary information and source data file. Source data are provided with this paper.

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

## Acknowledgements

Tissue was microdissected using equipment from CCBIO, Department of Clinical Medicine, The Gade Laboratory for Pathology, University of Bergen. LC-MS of discovery hypoxia experiments and microdissected tissue was performed at the Proteomics Unit at the University of Bergen. The authors would like to thank Mr. Bendik Nordanger, Centre for Cancer Biomarkers CCBIO, University of Bergen, for excellent technical assistance. This work was partly supported by the Research Council of Norway through its Centres of Excellence funding scheme, project number 223250 (to L.A.A.), and by grants from University of Bergen and Helse Vest Research Fund (to L.A.A.). This work was funded in part by NIH/NIGMS grant R01 GM132129 (J.A.P.).

## Author contributions

L.A.A. and E.B. conceived the study; experimental design by L.A.A., E.B., K.F., and S.K.; cell culture experiments were performed by S.K. and V.A.; laser capture microdissection was performed by K.F. with assistance from L.A.A.; proteomics experiments were performed by S.K., K.F., E.B. and J.A.P.; computational analysis were performed by S.K., K.F., H.V. and D.K.; E.W. and K.F. performed data analysis and statistical analysis on patient cohorts; immunohistochemistry was performed by S.K., I.W. and L.A.A.; S.K., K.F. and L.A.A. wrote the manuscript with contributions from H.V., D.K. and E.W.; G.K. established the patient cohort with assistance from S.A.; the study was directed by L.A.A.; all authors read and approved the final manuscript.

## Funding

## Competing interests

The authors declare no competing interests.
