## [Peer Review File · Nature Communications]

Hypoxia induced responses are reflected in the stromal proteome and provide essential information for breast cancer stratificationReviewers' Comments:

Reviewer #1:

Remarks to the Author:

The authors present a study of breast cancer cell line secretomes and assessed its clinical relevance in tumor specimens. They found differences between basal and luminal cell line secretomes and the differences between normoxic and hypoxic conditions. Their findings suggest that basal cells have similar features to luminal cells under hypoxia. The authors then compare the secretome to clinical sample stroma, and show the clinical significance of a 33 protein signature. While some of these analyses are novel, the overall findings are not, and quite expected from very well established differences among breast cancer subtypes. I therefore don't find the manuscript suitable for publication in Nature Communications.

There are several major points that require improvement:

1. The use of two cell lines per subtype is far from representing differences among subtypes. There may be multiple reasons for differences between cell lines, and their subtype identity is only one of them. For example there could be major differences related to tumor stage, metastatic state, grade etc. To obtain a reliable signature of differences among subtypes, one has to examine multiple cell lines. In addition, the cells are grown with different media which may have major impact on the proteome and secretome. For example, differences in glucose, pyruvate, amino acid concentrations may all affect hypoxia-related pathways.
2. The identified differences between luminal and basal cell lines and tumors have been studied extensively. While potentially these were not examined in the secretome, their findings are not actually different from those found in cell lines and multiple studies of tumor samples.
3. The authors should present the overlapping proteins that are significantly higher in luminal hypoxia vs. normoxia with the proteins that are higher in basal normoxia vs. luminal normoxia. This will show whether indeed the basal-high proteins confer a hypoxia-like phenotype.
4. Paragraph from line 81 discusses the prediction of TF regulation. Such predictions work much more accurately in the analyses of mRNA levels, not proteins. In addition, TF targets have large overlaps, so it is not reliable to predict specific TF signatures. Such a discussion has to refer to specific targets in order to ensure the reliability of the analysis.
5. Line 99 the authors refer to TCA cycle proteins, however none of these are TCA cycle proteins per se. Cytosolic isoforms of TCA cycle proteins are not actively participating in the cycle and may be related to other processes as well.
6. The extraction of the 33 protein signature is not clear. If understood correctly the authors extracted the overlapping proteins between the cell line secretome and the stromal differences between subtypes, and excluded the tumor region differences. I do not follow this logic, since the secretome is extracted from the cancer cell lines and not stromal ones. Therefore I do not see a reason why a stromal signature would be better than a tumor one.
7. MaxQuant parameters should be clearly stated. 'Recommended' parameters, as indicated by the authors is not sufficiently detailed.
8. Tables should be given in excel/txt files and they should also add full protein and peptide tables.

Reviewer #2:

Remarks to the Author:

This manuscript "Hypoxia induced responses are reflected in the stromal proteome and provide essential information for breast cancer stratification" is very interesting and correctly places the role of hypoxia in the breast cancer tumor process.

For this article to be publishable there are still some questions that need to be answered widely by the authors.

Figure 1: May be the authors should add Normoxic "cell line" secretome on the figure to remove any

mistake.

Question regarding VEGF: what about the vascularization status in the breast cancer? The authors should raise this issue in the Introduction

Sentence 78: Where PLB2 is coming from?

Sentence 116: What does it mean "to be pre-conditioned to a hypoxic environment"?

Sentence 134: "VEGFA, as the only protein," what does it mean? The sentence is not clear?

Part sentence 223 to sentence 241: This part, although very interesting, is not very clear. The authors should better explain the use of the Connectivity Map. The choice of NRF2 is not clear either.

Discussion: I disagree with the first sentences of the Discussion. After 30 years of research and discoveries, followed by a Nobel Prize in the field of hypoxia, it is difficult to say "Not much is known about the diversity of hypoxia response...this is relevant in a clinical context". The authors should revise their copy and be a little less haughty about previous works.

Sentence 302: It is difficult to understand that NRF2 has no inhibitor at present while NRF2 stands out in the paragraph on "Gene expression proposes compounds with potential relevance to 33P-high breast cancer". The authors talk about monobenzone and apigenin. The authors should clarify this point.

This article is very interesting but lacks a more "explosive" ending. One feels like saying "and so what? We navigate between VEGF and NRF2 without really knowing which one stands out the most. A final figure could allow readers to better understand the significance of these results. What are all the pathways or processes that stand out?

It is not clear from these results what impact this will have on triple negative therapy. The authors should address this in the Discussion.

Reviewer #3:

Remarks to the Author:

This manuscript describes work with the goal to better define the role of hypoxia in breast cancer subtypes. A limited set of ~4 human breast cancer cell lines purported to represent broad "basal" and "luminal" subtypes are subjected to normoxic and hypoxic conditions, and proteins from the cell culture media are identified by shotgun proteomics. Not surprisingly, they find that the "basal" and "luminal" secreted proteins are different from one another, and different with normoxia and hypoxia. As expected, they find that the proteins that are conserved between the two broad groups are involved in angiogenesis and/or hypoxia, e.g., VEGF. They go on to perform laser capture microdissection of tumor and stroma from ~16 breast cancer patients and by comparing the proteins identified in stroma of these patients with the proteomics data obtained from the cell lines they derive a 33 protein "signature" that, when compared to publicly available RNA data from the Metabric study, correlates with patients outcome in univariate analysis.

This manuscript is very densely written with some of the key data in Supplemental sections, it has many weaknesses, including that it ignores previous work in the literature on these topics of hypoxia and stromal markers of survival, that many of the key results have been reported in previous publications spanning back ~10-15 years, importantly that there are no experiments to confirm the bioinformatic inferences presents, and concerns about rigor and reproducibility. On this last point selection of BCa cell lines is questionable (MDA231 is claudin low subtype BT474 is Her2+, and not basal and luminal), there is no information on markers showing that stroma and tumor were separated

by microdissection (eg E-cad or vimentin expression), no power calculations/statistics to show 16 human samples will give meaningful results, etc.).

The work is not suitable for the high standard set by this journal.

Reviewer #4:

Remarks to the Author:

The paper of X and Y entitled "Hypoxia induced responses are reflected in the stromal proteome and provide essential information for breast cancer stratification" studies the proteins that are secreted in cell lines of luminal and basal nature, (1877) proteins altogether. In the first part of the study the authors compare the protein lists in the two types of cell lines, each grown in normoxic and hypoxic conditions. This leads to a number of comparisons (two types of cell lines, grown in 2 different conditions), which reveal that different proteins are secreted in each 4 situations (luminal/basal like and normoxic/hypoxic). I like Supplementary figure 1 and propose to bring it forward in the paper as well as build the presentation of the data according to where on the workflow one is. GSEA was performed to identify the biological functions, and they were found strikingly different in luminal vs basal, i.e. hypoxia triggers different signaling and metabolic pathways in both subtypes of BC. Only 7 proteins overlapped and showed increased abundance in both subtypes after hypoxia.

Then the authors performed the same proteomic analysis on laser captured microdissection specimens from breast cancer and identified 4,157 proteins In the epithelial tumor cell compartment, and 2,150 proteins in stromal samples. Of these 33 gene overlapped with the genes found from the cell lines experiment.

Then the authors "reverse-translate" these to mRNA and start interrogating larger mRNA compendiums (Metabric) using IPA to identify enrichment of genetic pathways and see if they can recapitulate the differential expression of these genes in the corresponding PAM50 subclasses and whether they bring difference in clinical presentation (survival).

The proteomics part of the paper (cell lines and microdissection) is rather straightforward and brings novel insight. I would suggest in this part the authors to separate better what is expected and confirmatory from what was known from before:

The proteins that separate luminal from basal-line- are any of them they part of PAM50?

The proteins that separate the stromal from the epithelial: how many are known from before?

The proteins that separate normoxic from hypoxic: how many are known from before and part of known and widely signatures of hypoxia?

When one performs correlation to hypoxia signatures, has one checked that the latter do not consist of the same genes?

Of the genes comprising the PAM50- how many proteins are in the epithelial and how many in the secreted compartment?

My main criticism is to the last part of the paper, the validation in Metabric and related cohorts. While it is a valid argument and valuable finding that the 33 gene signature is associated to survival, altogether and in LumA, LumB and Basal separately as well as related clinic-histo-pathological parameters (tumor size, grade), one could do a lot more than apply IPA to identify signaling and metabolic circuits that encompass these 33 proteins (see methods like Bayesian network inference, SEEK, etc) as well as machine learning methods, which one can employ which other pathways and genes are generated using these 3 as a seed. This may answer questions like where these proteins are produced, what cell types etc.

In fact at this stage of the analyses it is not clear why the authors want to restrict themselves to the 33 proteins, which represent the significant and unique differences between the subtypes in the

stromal compartment? I.e. if one looks for validation in bulk mRNA expression data that comprise of all cells admixtures, why restrict oneself to the 33 proteins that are found in the stroma and one does not even know where they come from?

Here one may go back and up the steps one came down to generate the 33 gene list and perform the same validation in Metabric (luminal vs basal, stromal vs epithelial, hypoxic vs normoxic).

Another critical moment is what the interaction to treatment response is and what part of the poor survival can be associated with poor treatment response. There are a number of datasets out there with known treatment, where one can look for interaction. produced? Perhaps with methods such as Cibersort or the like, one could see if this signature does not correlate to any cell type?

In conclusion, I like the clear thought line in this paper and the analytic workflow, but I think that in the synthetic part, when thinking of what the results mean, the authors can use more of the data they have generated and not only the 33 genelist. The advantage of such a defined and limited number of genes is in the possibility to assess the state of hypoxia non-invasively. Is there any evidence that any of these proteins are also detectible in serum or plasma?

RE: NCOMMS-21-47819

Hypoxia induced responses are reflected in the stromal proteome and provide essential information for breast cancer stratification

RESPONSE TO INDIVIDUAL REVIEWERS' COMMENTS

Reviewer #1, expertise in breast cancer proteomics (Remarks to the Author):

The authors present a study of breast cancer cell line secretomes and assessed its clinical relevance in tumor specimens. They found differences between basal and luminal cell line secretomes and the differences between normoxic and hypoxic conditions. Their findings suggest that basal cells have similar features to luminal cells under hypoxia. The authors then compare the secretome to clinical sample stroma, and show the clinical significance of a 33 protein signature. While some of these analyses are novel, the overall findings are not, and quite expected from very well established differences among breast cancer subtypes. I therefore don't find the manuscript suitable for publication in Nature Communications.

There are several major points that require improvement.

1.0. RESPONSE TO GENERAL COMMENTS:

We thank the reviewer for these comments. In general, we still believe that our approach and main findings are novel, by studying the hypoxic response in breast cancer cells (secretome) at the global proteomic level, by combining these findings with proteomic data derived from laser microdissected breast cancer stroma samples, and by identifying a 33P signature with strong clinical validation in terms of treatment-relevant patient stratification.

Further, our 33P signature does not only reflect or cover the known and much reported difference between luminal-like and basal-like subtypes, but was found to have clinical value beyond these phenotypes by independent prognostic value in multivariate survival analysis (whereas the PAM50 score was not significant). Overall, the main finding, that a proteomic stromal signature reflecting hypoxic stress was found to have clinical value beyond the traditional subtypes, represents a clearly novel finding, in our opinion, and could have strong practical value. We have now also added data indicating that 33P might have an interaction with radiation treatment, which could attract clinical interest.

1.1 COMMENT:

The use of two cell lines per subtype is far from representing differences among subtypes. There may be multiple reasons for differences between cell lines, and their subtype identity is only one of them. For example there could be major differences related to tumor stage, metastatic state, grade etc. To obtain a reliable signature of differences among subtypes, one has to examine multiple cell lines. In addition, the cells are grown with different media which may have major impact on the proteome and secretome. For example, differences in glucose, pyruvate, amino acid concentrations may all affect hypoxia-related pathways.

1.1. RESPONSE:

We appreciate the comments on this important issue, and we totally agree with the reviewer that choosing sufficient number of “representative” cell lines impacts the generalization of findings in breast cancer research. Integrated omics analyses from independent studies convincingly demonstrate that breast cancer cell lines closely resemble many molecular features and properties of breast cancers seen in the clinic. Such results provide additional evidence for selecting representative cell line models for signature discoveries. This is in agreement with our overall experimental design and validations presented so far, in particular the novel approach of using an integration of cell line secretome data (n=4) with microdissected breast cancer stromal proteome data (n=24), as well as transcriptomic data from METABRIC patients for validation.

We agree that the use of two cell lines per subtype is not representing the complete spectrum of characteristics among various available cell lines and clinical samples representing the two subtypes. Also, as pointed out by the reviewer, there may be multiple reasons for differences between cell lines, for example related to the characteristics of the clinical lesions from which they were derived (primary/metastatic, grade, receptor status, etc). To strengthen our work, we have therefore, as outlined below, included 8 additional cell lines (12 in total), and we have performed substantial revisions of the manuscript to strengthen the validity of our signature discovery approach.

Notably, it might be difficult to obtain “one signature” that captures all variations found in breast cancer cell lines and in particular in clinical samples, as there might potentially be several signatures representing different parts of the underlying biology and differences between luminal-like and basal-like phenotypes, as also pointed out by reviewer #4 (response 4.7).

An important point would be whether the discovery-based findings from a given panel of cell lines can be validated clinically. We would like to argue that we still consider our findings from these four initial cell lines, and their secretomes, to have been strongly validated by our microdissected breast cancer proteome data, showing distinct separation of basal-like and luminal-like subgroups in clinical cohorts, and supporting the reliability and validity of the 33P signature. Our findings when using the METABRIC and KMplotter cohorts convincingly demonstrate distinct differences between luminal-like and basal-like categories, even when using mRNA proxy data derived from whole tissue analyses. However, our findings would not exclude the value of other possible signatures. As we have outlined, the prognostic impact was even found to go beyond the known luminal/basal difference.

To some extent, the comments relate to the use of established cell lines as valid models of breast cancer subtypes, since the cell lines might reflect other differences than just the subtype. Still, we agree that expanding the number of cell lines would strengthen our approach, and we have added 8 additional cell lines (4 basal, 4 luminal).

In the revised manuscript, we have done the following (see details below), at three levels. #1. We present more clearly (in the revised Methods section, and Supplementary Methods) the arguments used for including the four original cell lines in the discovery phase of our study, and subsequently the additional 8 cell lines. #2.

We have done supplementary bioinformatic analyses to demonstrate the “representativity” and “positioning” of the four original cell lines, in comparison with the CCLE collection of breast cancer cell lines, both at protein and mRNA levels, as well as for selecting the 8 additional cell lines. #3. We have performed new experiments with 8 additional cell lines under hypoxic conditions to further support our data. The new hypoxia cell line experiment thus included the 4 initial cell lines and the 8 additional lines.

Part #1: Literature.

We performed extensive literature mapping to characterize available breast cancer cell line properties. By combining existing information with in-house bioinformatics analyses (below), we provide stronger evidence on the molecular “suitability” of specific “candidate” cell lines, to further support the selection of the initial four cell lines used in the “discovery phase” of our work. This information is summarized in **Supplementary Table 1** in the revised manuscript. The revised manuscript text has been amended to reflect the above-mentioned information.

The initial selection of four breast cancer cell lines was based on our aim to compare luminal-like and basal-like categories of breast cancer, their hypoxia responses, and the alterations reflected and detected in microdissected tumor stroma (tumor microenvironment). We intended to include frequently applied and well-established cell lines used by recognized teams and being well represented in the BC literature.

Briefly, for the luminal-like category, one “luminal A” and one “luminal B” cell line was selected, as well as one primary and one metastatic cell line. For the basal-like category, one primary and one metastatic cell line was selected; both of these were triple negative, and both were of the “basal B” type, see **Supplementary Table 1**, where we also include the 8 additional cell lines that we have now studied.

The (initially) selected luminal-like cell lines were both ER and PR positive. **BT-474** is derived from a primary invasive ductal carcinoma, expresses HER2 and is classified as luminal B (**Supplementary Table 1**). **MCF7** is derived from metastatic invasive ductal carcinoma (pleural effusion) and is characterized as luminal A breast cancer subtype. The selected basal-like cell lines were both triple-negative. **Hs578T** is derived from a primary invasive ductal carcinoma, and **MDA-MB-231** is derived from a metastatic adenocarcinoma (pleural effusion). Both basal-like cell lines showed claudin-low characteristics and belong to the basal B category of cell lines.

We would argue that the selected cell lines are widely used and included in several large studies investigating breast cancer cells *in vitro*^{1, 2, 3, 4, 5, 6}.

Representativity: All the tumor subtypes are not reflected as separate subtypes in breast cancer cell lines. The luminal A and luminal B breast cancer subtypes are not as well reflected in cell lines as in tumors and are represented as one luminal category *in vitro*¹. However, in our initial cell line selection, we decided to include a luminal cell line displaying HER2 overexpression, as would be categorized as luminal B in tumors, to better cover the specter of luminal-like tumors. The basal-like tumor subtype is further divided into basal A and basal B in cell lines^{1, 2}, and both of our initial cell lines were of the basal B category, which is the basal cell line category most resembling the triple-negative and well established clinical classification. The basal B cell line category

display more mesenchymal features and characteristics of TNBCs, and is considered to be more distinct from luminal cell lines than basal A¹. The basal-like tumor subtype is categorized as both basal A and basal B *in vitro*, but the triple-negative clinical characteristics are more resembling the basal B group^{1,2}. Further, the HER2-enriched tumor subtype is grouped to the basal A category of cell lines, however annotated with information on HER2 status. Therefore, our initial selection of basal-like cell lines was chosen from the basal B category. Regarding the claudin-low characteristic displayed by the basal B cell lines, the claudin-low subtype in tumors is grouped to the basal B category of cell lines with triple-negative characteristics, and the claudin-low subtype is not reflected as a separate subtype in cell lines (response to reviewer 3 (comment 3.4)).

Supplementary Table 1: Selected cell lines (initial and additional):

Cell line	Subtype	Receptor status			Tumor type	Source	Literature
		ER	PR	HER2			
Initial selection (discovery panel)							
BT-474	Luminal (B)	+	+	+	IDC	PT	1, 2, 3, 4, 7
MCF 7	Luminal (A)	+	+	-	IDC	PE	1, 2, 3, 4, 7
Hs 578T	Basal B (claudin-low)	-	-	-	IDC	PT	1, 2, 3, 4, 7
MDA-MB-231	Basal B (claudin-low)	-	-	-	AC	PE	1, 2, 3, 4, 7
Additionally selected cell lines (validation panel)							
HCC1428	Luminal (A)	+	+	-	AC	PE	1, 2, 3, 7
T47D	Luminal (A)	+	+	-	IDC	PE	1, 2, 3, 4, 7
ZR751	Luminal (A)	+	-	-	IDC	AF	1, 2, 3, 7
ZR-75-30	Luminal (B)	+	-	+	IDC	AF	1, 2, 3, 7
MDA-MB-468	Basal A	-	-	-	AC	PE	1, 2, 3, 4, 7
HCC1143	Basal A	-	-	-	DC	PT	1, 2, 3, 7
HCC1187	Basal A	-	-	-	DC	PT	1, 2, 3, 7
BT-549	Basal B (claudin-low)	-	-	-	IDC	PT	1, 2, 7

AC: adenocarcinoma. AF: ascites fluid. DC: ductal carcinoma. ER: estrogen receptor. HER2: human epidermal growth factor receptor 2. IDC: invasive ductal carcinoma. PE: pleural effusion. PR: progesterone receptor. PT: primary tumor. For additional information, see Neve *et al.*¹, Dai *et al.*², Kao *et al.*³, Holliday *et al.*⁴, and Nusinow *et al.*⁷.

Part #2: Bioinformatics.

First, we performed an unbiased exploratory analysis using publicly available transcriptomic (n=54)⁸ and proteomic (n=28)⁷ data from CCLE. For both transcriptomic and proteomic datasets, we used as input the available gene expression and protein expression matrices. The cell lines were projected into the 2D space using multidimensional scaling (MDS). In brief, MDS positions the objects into 2D in a way that preserves the distance of the objects in the multidimensional space. Thus, it is a non-linear dimensionality reduction technique that is very useful for visualization and exploration of datasets. **Supplementary Figure 11** in the revised manuscript show the orientation of the available breast cancer cell lines in the 2D space from CCLE transcriptomic and proteomic datasets. It is apparent that the cell lines formed clusters, and the clusters were strongly driven by their molecular subtype identity. This information was used as a guide to assess differences in the expression profiles of the

available cell lines, and unbiasedly select new candidate cell lines to “cover” the observed 2D space. We believe that the original four cell lines were neither outliers nor expressing very different transcriptomic or proteomic profiles from all other cell lines. Instead, they were quite representative in the 2D subtype space, similar to the additional 8 cell lines that we subsequently selected.

The revised **Supplementary Information (Methods)** describes our bioinformatics methodology, and we have included **Supplementary Figure 11** in connection to extended cell line validation of our discovery experiments,

Part #3: Extension of experiments for support and validation.

Additional experiments were conducted, and the revised version of our work presents data from altogether 12 cell lines, using hypoxic conditions and secretome proteomics. Moreover, *in silico* and *in vitro* mapping of these 12 cell lines has been performed, and additional data have been included in the revised manuscript, also supporting the suitability of the four “main cell lines” originally used.

Regarding differences in media in the two groups of cell lines, we have taken this into account in our new experiments and selection of cell lines. In the additional cell lines (validation panel, n=8), there is a balance between media and supplements in luminal-like and basal-like cell lines. In the originally selected cell lines, we have kept the same media in follow-up experiments. The hypoxia vs. normoxia analysis is not affected by differences in media across subtypes as the analyses are performed within each subtype and/or cell line. Additionally, some of the cell lines are grown in media not completely in line with ATCC standard recommendations, in which the cells have been adapted to the growth medium after consulting ATCC and following their recommended protocol for medium adaptation. In the case of MDA-MB-231 and MDA-MB-468, media adaptation was required as the cells were recommended grown in airtight cell culture flasks, which do not allow gas exchange needed for hypoxia exposure using a hypoxic chamber.

Expanding the cell line panel of **luminal-like cell lines**: we decided to include a HER2-positive cell line consistent with the luminal B tumor subtype, and three cell lines with hormone receptor status patterns corresponding to luminal A tumors. Importantly, regarding the HER2-positive cell lines included in our study (initial: BT-474; additional: ZR-75-30): these cell lines are hormone receptor positive and have luminal characteristics and belonging to the luminal category of cell lines (response to reviewer #3: comment 3.4).

Expanding the **basal-like category** in our cell line panel: three basal A cell lines and one basal B and claudin-low cell line were included to also have a balance between basal A and basal B cell lines in follow-up experiments to validate the results from the discovery phase. Importantly, all six basal-like cell lines were triple-negative. A complete overview of the initially and additionally selected breast cancer cell lines is presented in **Supplementary Table 1**. The basal A cell lines were included as this category is closely correlated with the basal-like tumor subtype^{9, 10}, and the basal B category of cell lines were included since these are more similar to the triple-negative tumors (see above).

All selected cell lines, both those initially selected and additional cell lines, are part of at least one of ATCC's cell line panels for breast cancer or triple-negative breast cancers, and none of the included cell lines are among the cell lines with debated subtype or characteristics (e.g., SKBR3, classified as luminal^{1, 3}, and later classified as HER2-enriched²).

The extended cell line validation experiment has been added to the revised manuscript, **Results page 16-17**:

“Validation of 33P by expanded cell line experiments. To validate the 33P signature derived from the original 4 cell lines (2 luminal-like, 2 basal-like), we added 8 additional cell lines (4 luminal-like, 4 basal-like) in a new validation experiment that included all 12 cell lines (see **Supplementary Information** for details, including **Supplementary Figure 11**) (see also **Figure 1d**). We predicted that this expansion would implicate a wider biologic spectrum with increased diversity and better coverage of hypoxia responses. First, we performed a discovery analysis on the validation experiment (12 cell lines), similar to the initial discovery, which resulted in a set of 36 proteins (36P) that correlated significantly with the 33P signature (correlation coefficient 0.70; $p < 0.001$; Pearson) (**Supplementary Figure 12**); the correlation was significant also when overlapping proteins ($n=10$) were omitted from 36P (correlation coefficient 0.50; $p < 0.001$; Pearson).

Next, we investigated the expression of the 33P proteins in the new dataset (12 cell lines) and found (by GSEA) that the 33P proteins were significantly associated with hypoxia (**Supplementary Figure 13**). Moreover, when examining individual proteins from 33P in the validation dataset, 13 of these proteins were still significantly altered in either the luminal-like or basal-like cell lines. A subscore consisting of these 13 proteins (13P) was generated in the METABRIC-Discovery cohort (mRNA data), showing significant association with tumor subtypes and patient survival among all luminal-like or basal-like breast cancer cases ($n=852$), as well as in the luminal A category ($n=466$) (**Supplementary Figure 14**). This result is similar to what was observed in the initial proteomic discovery data (resulting in 33P), indicating that the 13P subsignature represents a consistent set of hypoxia-altered proteins across the entire and expanded cell line panel ($n=12$). Notably, 13P was slightly stronger than 33P when directly compared in a multivariate analysis of patient survival (METABRIC-Discovery) (**Supplementary Table 13**), in particular among luminal A cases, indicating that the 13P subsignature is capturing a broader range of hypoxia responses and might reflect a wider set of aggressive stromal characteristics. The 13P signature score (Q4, upper quartile) was also associated with reduced survival in the KMplotter dataset (**Supplementary Figure 14**), and 13P was significantly associated with response to radiation therapy ($p=0.035$, test for interaction). Taken together, these data support the validity of the original 33P signature. At the same time, the 13P signature, based on the extended experiments, showed slightly stronger prognostic impact.”

1.2. COMMENT:

The identified differences between luminal and basal cell lines and tumors have been studied extensively. While potentially these were not examined in the secretome, their findings are not actually different from those found in cell lines and multiple studies of tumor samples.

1.2. RESPONSE:

We thank the reviewer for this comment. Given our novel approach, we see it as a strength and validation of our data that parts of the results are in line with previously published work.

The integrated interrogation done by combining data on the effects of hypoxia on breast cancer cell secretomes with microdissected patient material is, to our knowledge, a novel approach. We have identified a potential signature biomarker for breast cancer disease progression and patient stratification, and our signature (33P) was validated in clinical cohorts and found to be stronger than the subtype difference (based on PAM50), with independent prognostic value. Also, 33P might be a potential predictive marker for radiation therapy.

The finding of a proteomic stromal signature, reflecting tumor hypoxia, and providing strongly validated and treatment-relevant clinical information beyond what is currently available, is in our opinion a novel concept. These data might attract clinical interest, in particular related to the independent prognostic split of low-grade luminal cases. For further distinguishing the novel and known parts of our results, this is described in more detail under reviewer 4, comments 4.1 to 4.5.

1.3. COMMENT:

The authors should present the overlapping proteins that are significantly higher in luminal hypoxia vs. normoxia with the proteins that are higher in basal normoxia vs. luminal normoxia. This will show whether indeed the basal-high proteins confer a hypoxia-like phenotype.

1.3. RESPONSE:

We thank the reviewer for this comment and have investigated the overlapping proteins in our data sets and found 37 proteins in common for the proteins with higher secretion from basal-like cells at baseline and proteins with hypoxia-increased secretion in luminal-like cell lines. We would like to point out that the proteins within this overlap have been of our interest in relation to angiogenesis, especially when discovering this pattern for VEGFA, which is a direct target of HIF1A and a secreted marker for hypoxia, as well as considered one of the main angiogenic factors¹¹. Additionally, of the HIF1A targets (from IPA analysis of the 150 hypoxia-upregulated proteins), 6 of the 10 HIF1A targets in the luminal-like hypoxia response proteins are among these 37 overlapping proteins (LDHA, EIF5A, GAPDH, NPM1, VEGFA, HSPA4), including proteins of interest in this context (VEGFA, LDHA). The manuscript text has been updated where these proteins are relevant, and all proteins have been marked in **Supplementary Table 3** edited with this additional information.

Revised manuscript text:

"(...) several of the basal-like angiogenic proteins were already higher at baseline, compared with hypoxia-increased luminal-like angiogenic proteins, including ANG, NCL, PRCP and VEGFA (**Supplementary Table 3**)."

1.4. COMMENT:

Paragraph from line 81 discusses the prediction of TF regulation. Such predictions work much more accurately in the analyses of mRNA levels, not proteins. In addition, TF targets have large overlaps, so it is not reliable to predict specific TF signatures. Such a discussion has to refer to specific targets in order to ensure the reliability of the analysis.

1.4. RESPONSE:

We agree that prediction of TF regulation can more accurately be read from gene expression changes, measured at the mRNA level; however, investigation of changes in gene expression would require analysis of the cell lysates of the breast cancer cell lines exposed to hypoxia, and longer exposure time to hypoxia than 24h, which is outside the focus of this manuscript.

The rationale for using Ingenuity Pathway Analysis (IPA, Qiagen) as a tool to predict upstream regulators: In this study we aimed to investigate breast cancer cells' initial response to hypoxia that would happen within a short time window (24h). We would expect to see differences in TFs contributing to this change. Especially, HIF1A is associated with acute hypoxia response (cell culture hypoxia exposure less than 24 hours), whereas HIF2A is stabilized in chronic hypoxia. Further, we also wanted to predict other TFs potentially involved in regulating the expression of these secreted proteins. For this purpose, we used IPA, which is a functional analysis software that can analyze gene expression as well as protein abundance patterns using a build-in scientific literature database (QIAGEN Knowledge Base according to: digitalinsights.qiagen.com). This has been clarified in the revised manuscript.

Based on the secreted proteins with increased abundance in the media of breast cancer cell lines exposed to hypoxia, we used software tools to predict potential upstream regulators of the initial secretome response found. We expect the predicted TFs to be expressed in breast tumors as a result of hypoxic conditions.

With regards to the comment on overlapping TF targets, we analyzed both combined and separately the luminal-like and basal-like hypoxia responses (proteins with increased abundance), we expect that potential upstream regulators of luminal-like and basal-like hypoxic secretomes might be shared, as these are transcriptional regulators that work upstream of the secreted proteins, as stated in the text. Notably, we identified NRF2 as distinct for the luminal-like hypoxic secretome, while TFEB and BCL6B were predicted to lie upstream of the basal-like hypoxic secretome.

In connection to the results from this analysis, we have now assessed the expression of predicted upstream regulator NRF2 in breast tumor tissue by IHC, and results have been included in the revised version of the manuscript (see **Supplementary Figure 10** for IHC of NRF2 in breast cancer tissue from our breast cancer cohort, n=42, in response to comment 3.3).

1.5. COMMENT:

Line 99 the authors refer to TCA cycle proteins, however none of these are TCA cycle proteins per se. Cytosolic isoforms of TCA cycle proteins are not actively participating in the cycle and may be related to other processes as well.

1.5. RESPONSE:

We thank the reviewer for this comment. It is true that MDH1 and IDH1 are cytoplasmic, and therefore not a part of the textbook representation of the TCA cycle. For our enrichment studies, we have applied the widely used and curated “Biological processes” database from “Gene ontology”, collected from the molecular signatures database (MSigDB). In the gene set with the provided gene ontology identifier (GOID 6099), these cytoplasmic proteins are included as members of the TCA cycle. The cytoplasmic isozymes catalyze the same reaction as the mitochondrial isozymes, and while not actively participating in the cycle, they seem to be involved in regulation of the TCA cycle. Still, we have tried to be careful by stating that these proteins are “...associated with metabolic processes such as (...) TCA cycle” to avoid misunderstandings.

1.6. COMMENT:

The extraction of the 33 protein signature is not clear. If understood correctly the authors extracted the overlapping proteins between the cell line secretome and the stromal differences between subtypes, and excluded the tumor region differences. I do not follow this logic, since the secretome is extracted from the cancer cell lines and not stromal ones. Therefore I do not see a reason why a stromal signature would be better than a tumor one.

1.6. RESPONSE:

We would like to thank the reviewer for this comment. First, we propose to include a schematic overview of the signature protein extraction in previous **Supplementary Figure 1 (now main Figure 1, panel c, in response to comment 4.0)** to improve clarity of the various steps in our approach.

Our focus has been on secreted proteins, from tumor cells to the surrounding stroma (or TME), assuming that this is correlated to functional properties and roles of these proteins, in developing aggressive characteristics of the tumor microenvironment. Therefore, we asked whether secreted proteins from the cell lines could be found in tumor stroma from clinical samples, and whether the subtype contrast in this overlapping set of proteins, indicating the hypoxic response, could reflect “aggressive stromal features or responses” that would relate to important clinical and prognostic features. Whether a stromal proteome signature (reflecting hypoxia) would compare and compete with tumor cell-based features (such as PAM50 based subtypes) represented one of our research questions, and we found an independent value of the stromal proteome signature (33P) beyond the tumor cell-based subtype difference (PAM50 based). The manuscript text has been updated to improve clarity, including a new paragraph in the methods section on signature discovery:

The new paragraph is as follows:

“Signature discovery. The signature proteins were derived from integrated analysis of secretomes from discovery BCCL and microdissected stromal tissue proteomics data. The proteins that were in common for the hypoxia-increased proteins (hypoxia vs. normoxia) and the stroma-exclusive subtype differences (basal-like vs. luminal-like) were extracted as the protein signature (see **Figure 1c**). The signature proteins were validated in the extended validation panel of BCCL.”

1.7. COMMENT:

MaxQuant parameters should be clearly stated. 'Recommended' parameters, as indicated by the authors is not sufficiently detailed.

1.7. RESPONSE:

We agree, and we have now edited the manuscript text to the following:

“Computational analysis of proteomics data. Raw MS files from discovery BCCL panel secretomes and microdissected tissues were analyzed using MaxQuant¹² (v1.5.3.30 for conditioned medium samples and v1.6.0.16 for patient samples) with label-free quantification and “match between runs” enabled. The precursor ion tolerance for total protein level profiling was set to 20pmm, and product ion tolerance to 0.5 Da. Carbamidomethylation of cysteines was set as fixed modifications, and oxidation of methionines and N-terminal acetylation was set as variable modifications. False discovery rate (FDR) for peptide and protein identification was set to 1%. MS/MS spectra were searched in the Andromeda search engine against the forward and reverse Human UniProt database.”

1.8. COMMENT:

Tables should be given in excel/txt files and they should also add full protein and peptide tables.

1.8. RESPONSE:

We agree, and this has now been performed. Attached is an Excel file containing the supplementary tables as also presented in the Supplementary Information document. Additional tables with full protein and peptide information from hypoxia experiments on cell lines, discovery and validation panels have been provided in the Supplementary Data file (source data).

Reviewer #2, expertise in hypoxia (Remarks to the Author):

This manuscript "Hypoxia induced responses are reflected in the stromal proteome and provide essential information for breast cancer stratification" is very interesting and correctly places the role of hypoxia in the breast cancer tumor process. For this article to be publishable there are still some questions that need to be answered widely by the authors.

2.1. COMMENT:

Figure 1: May be the authors should add Normoxic “cell line” secretome on the figure to remove any mistake.

2.1. RESPONSE:

We would like to thank the reviewer for the general comments, as well as for the specific comment #2.1 pointing this out. We have updated the figure title and figure legend to avoid any misunderstanding.

Figure 2: Distribution of secreted proteins between oxygen conditions and subtypes in breast cancer cell secretomes.

Relative distribution of secreted proteins between the luminal-like and basal-like cell lines under normoxic (a) and hypoxic (b) conditions, and between the normoxic and hypoxic conditions from luminal-like (c) and basal-like cell lines (d). Colored circles represent significantly differentially secreted proteins between the comparisons (Student's T-test, $p < 0.05$).

2.2. COMMENT:

Question regarding VEGF: what about the vascularization status in the breast cancer? The authors should raise this issue in the Introduction

2.2. RESPONSE:

Thank you for this comment. There is clearly a relationship between tissue hypoxia and vascularization, and we have now mentioned this aspect in the Introduction part.

Our data already indicate this positive association, as demonstrated for example in **Supplementary Figure 5 i-j**. Also, we have explored our data further and find that 33P (by MS-proteomics) is positively associated with vascular proliferation ($n=42$), as an indication of activated angiogenesis ($p=0.05$). This has been included in the revised manuscript.

2.3. COMMENT:

Sentence 78: Where PLB2 is coming from?

2.3. RESPONSE:

We apologize for the lack of clarity, and we agree with the reviewer that additional explanation is indicated for PLBD2. We have now included this information on PLBD2 in the Results section. Our intention was to point out that this protein has not previously been associated with any cancer, and follow-up studies would therefore be indicated.

The updated text in the Results section is as follows:

“Next, using the IPA database combined with literature mining, we found that of the 150 hypoxia-increased secreted proteins, Putative phospholipase B-like 2 (PLBD2) have not been previously associated with cancer. Based on sequence similarity,

PLBD2 is a putative phospholipase, and probably involved in fatty acid metabolism. Studies are needed to elucidate the role of PLBD2 in cancer.”

2.4. COMMENT:

Sentence 116: What does it mean “to be pre-conditioned to a hypoxic environment”?

2.4. RESPONSE:

We apologize for the lack of clarity. For certain hypoxia-associated proteins, including VEGFA and LDHA, we observed a higher baseline level of secretion from basal-like cell lines, than induced by hypoxia in luminal-like cell lines (also see comment 1.3 and response). We hypothesized that the higher levels of these proteins may be due to a phenotype in the basal-like cells (at baseline) that is similar to changes induced by hypoxia in luminal-like cells.

We have rewritten the relevant paragraph to clarify. Thematically related to this comment, rephrasing of the paragraph on VEGFA is presented under comment 2.5, regarding the VEGFA protein, and angiogenic proteins under comment 1.3.

The updated text is as follows:

“Whereas we did not observe significant hypoxia-induced differences in energy metabolism among basal-like cells, these cell lines still showed 1.9-fold higher levels of LDHA at normoxia compared to the luminal-like hypoxic secretome ($p=0.002$). In contrast, the basal-like hypoxome showed enrichment related to tissue development, immune responses, inflammation and secretion (**Supplementary Table 8**). Taken together, this suggests that luminal-like cells have a stronger hypoxia response, while the basal-like cells may have already adapted to a hypoxic environment *in vivo*, as hypoxic and necrotic regions are more frequently present in rapidly growing tumors, such as basal-like breast cancers.”

2.5. COMMENT:

Sentence 134: “VEGFA, as the only protein,” what does it mean? The sentence is not clear?

2.5. RESPONSE:

We apologize for the lack of clarity. We agree with the reviewer that this sentence should be improved, and we have made changes in the text to clarify.

The updated text is as follows:

“Notably, among the 150 hypoxia-increased proteins, only 8 were associated with angiogenesis (**Table 1**). Vascular endothelial growth factor A (VEGFA) was the only angiogenesis-related protein that showed increased secretion under hypoxic conditions in both luminal-like and basal-like breast cancer cells, as compared to normoxic conditions. VEGFA showed 3.7-fold higher abundance in normoxic basal-like secretomes compared with hypoxic luminal-like secretomes ($p=0.01$), and this difference was validated by enzyme-linked immunosorbent assay (ELISA) (**Supplementary Figure 2**).”

2.6. COMMENT:

Part sentence 223 to sentence 241: This part, although very interesting, is not very clear. The authors should better explain the use of the Connectivity Map. The choice of NRF2 is not clear either.

2.6. RESPONSE:

We apologize for the lack of clarity, and we would like to thank the reviewer for the comment. We have modified the text with clarifying statements. We would like to point out that the Connectivity map analysis is used as a hypothesis-generating tool to point towards possible use of our signature in a clinical context.

Our focus on NRF2 from the CMap analysis is additionally based on findings from IPA analysis, in which we find NRF2 as one of the top-ranked upstream transcription factors for the luminal-like hypoxia response from our secretome data.

Interestingly, from the IPA analysis, we would have expected NRF2 to be inhibited by the CMap compounds, however, the top-ranked CMap compounds suggest antioxidant effects and/or promoting the NRF2 activity (apigenin).

NRF2 was selected for follow-up experiments by IHC (response to comment 3.3). Interestingly, we found a positive correlation between stromal staining intensity and the 33P signature scores in these patients, a pattern in line with NRF2 results from the IPA analysis. These results have been presented in a new paragraph (presented under comment 3.3) and the manuscript text regarding CMap analysis has been modified.

See also response to comment 2.8.

The manuscript text has been edited to the following:

“Gene expression propose compounds with potential relevance to 33P-high breast cancer. To search for biologically relevant targets in 33P-high breast cancer, we queried the drug signature database Connectivity Map (version 02)¹³ for compound-related gene expression profiles negatively enriched in 33P-high tumors, as these compounds may contribute to decrease some of the features associated with high 33P signature scores. Among 1,309 small molecules represented in Connectivity Map, expression profiles from compounds with properties promoting attenuation of tumor effects from hypoxia were top ranked (**Supplementary Table 12**). Previous studies on many of these compounds have demonstrated anti-hypoxia effects in cancer (e.g., resveratrol¹⁴, sirolimus¹⁵). Several of the top-ranked compounds have also been demonstrated to have effects on oxidative stress and/or the transcription factor NRF2 (nuclear factor erythroid 2-related factor 2), which is encoded by the NFE2L2 gene (e.g., apigenin¹⁶). NRF2, which was also found in our IPA analysis of upstream transcription factors for luminal-like hypoxia response proteins, is a known regulator of genes containing antioxidant response elements^{17, 18}.

In stratified CMAP analyses (luminal-like and basal-like separately), gene expression profiles of compounds with PI3K/mTOR inhibitory properties were top-ranked and negatively enriched in 33P-high tumors (**Supplementary Table 12**). Adding to this, signatures reflecting PI3K/AKT/mTOR activation were top-ranked and significantly

enriched in tumors with high 33P (mRNA) score (GSEA/MSigDB; H and C6 subsets; FDR<0.05). Taken together, results from the CMAP analyses, used as a hypothesis-generating/supporting tool, propose a biological relevance of NRF2 activating and/or PI3K/mTOR inhibitory compounds to 33P-high tumors.”

2.7. COMMENT:

Discussion: I disagree with the first sentences of the Discussion. After 30 years of research and discoveries, followed by a Nobel Prize in the field of hypoxia, it is difficult to say "Not much is known about the diversity of hypoxia response....this is relevant in a clinical context". The authors should revise their copy and be a little less haughty about previous works.

2.7. RESPONSE:

Thank you for this comment, and we apologize for the lack of clarity. We clearly agree that there is a huge amount of literature on hypoxia responses. The point we were trying to make was that most of the hypoxia research has focused on hypoxia-inducible factors (HIFs) and their target genes (including the work highlighted by the Nobel prize winners in 2019^{19, 20, 21, 22, 23}), but less is known about the diversity and heterogeneity of hypoxia responses in clinical samples and at the global proteomic level, and whether this can be linked to clinical correlates such as disease progression. Our hypothesis, underlying this work, has been that stromal proteomic profiles, reflecting hypoxia, could potentially demonstrate information beyond what is previously observed and captured by traditional subtypes (e.g. based on PAM50 transcriptomics). Still, there are many unknowns in the field of hypoxia, especially in tumor hypoxia.

The first part of the Discussion has been rephrased and expanded to the following:

“Hypoxia appears to be a master driver of tumor progression^{24, 25}, and extensive work has been performed on hypoxia-inducible factors (HIFs) and their target genes. However, less is known about the hypoxia responses at the global proteomic level in cancer, and whether these are relevant in a clinical context. Although hypoxia has been investigated using MS-based proteomics, the main focus has been on cellular proteins and pathways²⁶. Even with an increasing number of cancer secretome studies, also including hypoxic secretomes^{27, 28}, the majority of these have concentrated on extracellular vesicles and their role in metastasis^{29, 30}.”

2.8. COMMENT:

Sentence 302: It is difficult to understand that NRF2 has no inhibitor at present while NRF2 stands out in the paragraph on "Gene expression proposes compounds with potential relevance to 33P-high breast cancer". The authors talk about monobenzene and apigenin. The authors should clarify this point.

2.8. RESPONSE:

We apologize for the lack of clarity. We would like to thank the reviewer for pointing out this weakness for improvement and we have revised the manuscript text to clarify this point and supplemented with additional information.

Connectivity Map analysis is a data-driven hypothesis generating-tool, as used for our analysis, to predict compounds that can increase (positive enrichment) or decrease

(negative enrichment) features of gene expression patterns associated with the 33P signature. The gene expression patterns are generated from cell lines treated with these compounds. It is therefore possible that compounds have an effect in cell cultures and top-ranked in Connectivity Map analysis while not passing clinical trials and approved for treatment in patients.

The manuscript text has been edited to the following:

“NRF2, considered a master regulator of cellular antioxidant response¹⁷, was identified as an upstream regulator of luminal-like hypoxia responses (by IPA analysis), and stromal NRF2 protein expression by IHC was associated with high 33P in our breast cancer cohort. Using the CMAP database for drug response and repurposing exploration, NRF2 (by activation) and PI3K (by inhibition) were pointed out as potential targets in hypoxic tumors³¹. From our IPA analysis, we would expect inhibition of upstream transcription factors of our hypoxia-increased proteins, including NRF2, and it is currently not clear how these apparently paradoxical results might be explained. Regarding the PI3K/mTOR pathway, also indicated by our CMAP exploration, only few clinical trials have tested PI3K and/or mTOR inhibitors in advanced triple negative breast cancer, with published data from one trial so far (*i.e.*, NCT00499603), and with no change in response rate (at 12 weeks) following downregulation of mTOR³². Whether stratification by 33P would have provided more information, is not known.”

2.9. COMMENT:

This article is very interesting but lacks a more "explosive" ending. One feels like saying "and so what? We navigate between VEGF and NRF2 without really knowing which one stands out the most. A final figure could allow readers to better understand the significance of these results. What are all the pathways or processes that stand out?

2.9. RESPONSE:

In the revised manuscript, additional data is presented, validating the 33P both experimentally and clinically (see **Figure 1, panel c**) in terms of prognostic stratification beyond what is known. Of particular interest, it appears that 33P might potentially be a predictive marker for radiation therapy in breast cancer, in particular among luminal cases. This fits with the literature on hypoxia and effect of radiation treatment.

With respect to a final figure, such as a “graphical abstract”, we understand that this would not be in compliance with journal guidelines. If this is not correct, we would be willing to include this in our presentation.

2.10. COMMENT:

It is not clear from these results what impact this will have on triple negative therapy. The authors should address this in the Discussion.

2.10. RESPONSE:

Thank you for this comment. We agree with the reviewer that it is not clear how our results might impact the therapy of triple negative breast cancer. Maybe the most important finding relating to our clinical validation is the ability of the 33P signature to

split the luminal category, indicating that a subgroup of these patients might potentially benefit from additional therapy. The 33P as a novel “biomarker” to substratify luminal-like or luminal A breast cancer might possibly be highly relevant in a clinical context. Also, as mentioned above (#2.9), our new data points to a potential role of 33P as a predictive marker for radiation therapy.

Still, our data also indicate that the 33P could split the basal-like category (**Figure 5c; Supplementary Figure 7**). However, whether some tumors could have less intense therapy, or other tumors should receive additional therapy, is not clear.

An important finding of clinical relevance is that the 33P might represent a novel “signature biomarker” that could potentially improve patient stratification.

Reviewer #3, expertise in breast cancer TME (Remarks to the Author):

3.1. COMMENTS:

This manuscript describes work with the goal to better define the role of hypoxia in breast cancer subtypes. A limited set of ~4 human breast cancer cell lines purported to represent broad “basal” and “luminal” subtypes are subjected to normoxic and hypoxic conditions, and proteins from the cell culture media are identified by shotgun proteomics. Not surprisingly, they find that the “basal” and “luminal” secreted proteins are different from one another, and different with normoxia and hypoxia. As expected, they find that the proteins that are conserved between the two broad groups are involved in angiogenesis and/or hypoxia, e.g., VEGF. They go on to perform laser capture microdissection of tumor and stroma from ~16 breast cancer patients and by comparing the proteins identified in stroma of these patients with the proteomics data obtained from the cell lines they derive a 33 protein “signature” that, when compared to publicly available RNA data from the Metabric study, correlates with patients outcome in univariate analysis.

3.1. RESPONSE:

We thank the reviewer for these comments. The remark on the limited set of cell lines is covered in our previous response #1.1., with experimental extensions and additions in the manuscript.

Also, we would like to point out that the number of microdissected breast cancers is 24 (not 16), for proteomic interrogation.

Further, the value of the 33P signature was found to be significant by validation not only in univariate survival analysis, but also by multivariate modelling, thus indicating independent impact on survival, when adjusting for tumor size, lymph node metastasis, histologic grade and PAM50 (basal-like vs. luminal-like) subgroups, demonstrating prognostic value beyond current categories and with potential clinical significance. Also, as mentioned above (#2.9-10), our new data points to a potential role of 33P as a predictive marker for radiation therapy, as now reported in the revised manuscript.

3.2. COMMENTS:

This manuscript is very densely written with some of the key data in Supplemental sections, it has many weaknesses, including that it ignores previous work in the literature on these topics of hypoxia and stromal markers of survival, that many of the key results have been reported in previous publications spanning back ~10-15 years,

3.2. RESPONSE:

We thank the reviewer for this comment, and we apologize for the lack of clarity in relation to previous work. While the basic literature on hypoxia is well recognized, we will still argue that our approach is novel, by combining cell line “hypoxic” secretome with proteomic data from microdissected breast cancer stroma for potential clinical relevance. Moreover, the 33P signature emerging from this combined interrogation has been strongly validated in terms of clinical relevance. The fact that this hypoxia-related stromal signature carries independent prognostic significance beyond standard tumor features including the molecular subtype (based on PAM50) might be very important, and the ability of this proteomic stromal signature to substratify the luminal breast cancer categories is a novel and potentially very useful observation, in our opinion. Notably, since 33P is considered a hypoxia-related stromal signature, it is very interesting that a significant interaction with radiation treatment was found, and this has now been described in the revised manuscript (as mentioned above).

The literature on hypoxia for our research focus is now covered in more detail in comment 2.7, and in the revised manuscript. Among the hypoxia markers are expression of hypoxia-inducible factors (HIFs), associated with reduced overall and disease-free survival in breast cancer³³. Further, VEGFA, which is a direct target of HIF1A and a secreted marker for hypoxia³⁴, is associated with more aggressive features such as angiogenesis¹¹. Regarding stromal markers of survival, stromal expression of individual markers such as NGAL, CD68 and Syndecan-1, and the dual markers COX-2 and COL1A1 are associated with worse outcome in breast cancer patients³⁵. Information on the “global stromal proteome” related to hypoxia and reported in a set of clinical samples followed by strong clinical validation, has to our knowledge not been presented. More references have been added in the revised manuscript.

3.3. COMMENTS:

importantly that there are no experiments to confirm the bioinformatic inferences presents,

3.3. RESPONSE:

We appreciate this comment, and we have performed additional experiments for further investigations and validation of our bioinformatics results.

In addition to VEGFA, as presented in the originally submitted manuscript, we have performed enzyme-linked immunosorbent assay (ELISA) on angiopoietin-like 4 (ANGPTL4) and cathepsin B (CTSB). VEGFA is a secreted hypoxia marker and a direct target of HIF1A and was in addition to this selected as it showed significantly increased secretion in response to hypoxia in both luminal-like and basal-like breast cancer cell lines. Further, we selected ANGPTL4 to be evaluated by ELISA as a hypoxia-increased angiogenic protein for the basal-like hypoxia response only, as

gene set enrichment analysis (GSEA) found angiogenesis to be overrepresented in the basal-like compared to luminal-like secretomes at both baseline and under hypoxic conditions, and ELISA experiments validated this.

CTSB, which is also associated with angiogenesis³⁶, was selected due to hypoxia-increased secretion in both breast cancer subtypes as well as being part of the 33P signature proteins. When investigating the differences in CTSB secretion from the luminal-like or basal-like subtypes as a group, we did not find a significant difference in ELISA results. However, when investigating the secretion level within individual cell lines, we found patterns in line with the MS data. The luminal levels of CTSB are below the limit of detection (not detected; BT-474) and/or the assay range in ELISA and could not be accurately predicted. This is however consistent with the lower levels of secretion from luminal-like than basal-like cell lines.

Based on findings from IPA analysis predicting NRF2 as an upstream transcription factor for luminal-like hypoxia response, and CMAP analysis predicting benefit of NRF2-activating and/or antioxidant compounds, we selected NRF2 to be evaluated by immunohistochemistry. Here, we found a significant positive correlation between NRF2 stromal staining and 33P signature scores from the same patients.

The new results have been presented in a new paragraph:

“Immunohistochemical expression of NRF2 in tumor tissue. Based on results from IPA and CMAP analyses, IHC was performed to evaluate NRF2 expression in the tumor stromal and epithelial compartments using a breast cancer cohort of 42 cases with tissue proteomics information and 33P status. Stromal NRF2 expression (**Supplementary Figure 10**) was found to be significantly correlated to the 33P signature scores ($\rho=0.56$, $p<0.001$), supporting our IPA findings (above); however, epithelial NRF2 expression was not associated with 33P.”

The additional methods section for immunohistochemistry is as follows:

“Immunohistochemical staining. Immunohistochemistry detection of NRF2 expression in tissue samples was performed manually on 4-5 μm thick tissue microarray (TMA) sections from formalin-fixed paraffin-embedded tumor tissue from an in-house cohort of breast cancer patients ($n=42$; luminal-like 23, basal-like 19) with MS-proteomics information in parallel. Briefly, target retrieval for NRF2 was performed in Ventana Benchmark Ultra staining platform (Roche Tissue Diagnostics, Ventana Medical Systems, USA) with Cell Conditioning (CC1, # 06414575001, Roche Tissue Diagnostics, Ventana Medical Systems, USA) (pH9) at 95°C for 48 minutes before endogenous peroxidases were blocked with Inhibitor CM (from DAB-kit #5266645001, Roche Tissue Diagnostics, Ventana Medical systems) at 37°C 4 minutes. Slides were incubated with a monoclonal rabbit antibody against NRF2 (ab62352, Abcam, USA, diluted 1:100) for 60 minutes, followed by incubation with EnVision rabbit HRP (#K400311-2, Agilent, USA) for 30 minutes. To add color at the site of target antigen recognized by the primary antibody, DAB chromogen (#K346811-2, Agilent, USA) was applied for 10 minutes. Finally, sections were rinsed in distilled water and counterstained with Haematoxylin (#S330130-2, Agilent, USA).

NRF2 staining was recorded using a semi-quantitative and subjective grading system, considering the intensity of staining (none = 0, weak = 1, moderate = 2, and strong = 3) in tumor stromal and epithelial areas separately³⁷.”

ELISA results are incorporated in the existing figure (**Supplementary Figure 2**) together with corresponding plots of LFQ intensities from MS analysis.

3.4. COMMENTS:

and concerns about rigor and reproducibility. On this last point selection of BCa cell lines is questionable (MDA231 is claudin low subtype BT474 is Her2+, and not basal and luinal),

3.4. RESPONSE:

We thank the reviewer for these comments. The selection of cell lines is now covered in comment #1.1. We have expanded the cell line panel from 4 to 12 and now present arguments and results validating our approach and initial findings. ***We have included the additional points from the present comment #3.4 also in the response to comment #1.1.*** In the accompanying table (**Table 1 in this letter**) we present a listing and categorization of the 12 included breast cancer cell lines based on current literature, and this panel now covers a wider spectrum of cell lines with various characteristics. As for the BT474, we believe that the combination of HR+ and HER2+ corresponds to the luminal B tumors, and that the MDA-MB-231 claudin-low cell lines is part of the broader basal B cell line category, also in line with other investigations of these cell lines, as references in comment 1.1.

3.5. COMMENTS:

there is no information on markers showing that stroma and tumor were separated by microdissection (eg E-cad or vimentin expression),

3.5. RESPONSE:

We thank the reviewer for this valid comment, and we agree that it might be difficult to obtain complete purity when separating epithelial and stromal tumor compartments. We did not use a protein marker for *in situ* guidance of the laser microdissection. However, we used digital high-resolution images (0.2 μm per pixel) of parallel sections stained with hematoxylin-eosin with guidance of a very experienced breast pathologist (L.A.A). We have now clarified the text in the revised manuscript to increase clarity.

The updated manuscript text is as follows:

“Ten micrometer thick formalin-fixed paraffin-embedded (FFPE) sections were deparaffinized, rehydrated and stained with hematoxylin. Breast cancer epithelium and tumor stroma (adjacent non-epithelial tissue) were laser capture microdissected (PALM MicroBeam, Zeiss) and pressure catapulted into a tube cap (AdhesiveCap 500 opaque, Zeiss). Tumor epithelium and tumor stroma areas were selected under supervision of an experienced breast pathologist (L.A.A), using digital high-resolution images of parallel sections stained with hematoxylin-eosin. Depending on availability 0.5-1.9 $\times 10^7$ μm^3 tissue was obtained.

Subsequently, to estimate the amount of tumor epithelium “contamination” in the tumor stroma fractions and secure the quality of the separation, we compared the intensities of the epithelial marker cytokeratin-8 in the tumor epithelial and the tumor stroma samples after proteomics analysis (**Supplementary Figure 15**). We found on average 62-fold higher intensities of cytokeratin-8 in the tumor epithelium compared to the tumor stroma, respectively (basal-like: 68-fold, $p < 0.001$; luminal-like: 56-fold, $p < 0.001$). By estimation, on average, only 1.6% (median: 1.7%) epithelial tissues was present in the stromal samples. The low levels of epithelium in microdissected stroma was true for both basal-like and luminal-like samples; the luminal-like samples had on average 6.1-fold higher content of cytokeratin-8 compared to basal-like samples in tumor epithelium ($p < 0.001$). This was as expected since cytokeratin-8 is higher expressed in luminal compared with basal-like epithelial cells. We believe that these data validate the purity of our microdissected samples.”

3.6. COMMENTS:

no power calculations/statistics to show 16 human samples will give meaningful results, etc.).

3.6. RESPONSE:

Thank you for this comment. First, the number of cases with microdissected tumor stroma was 24, not 16.

As for power calculations upfront, we believe that these are typically performed in the setting of randomized clinical trials (RCT), to calculate the needed number of patients enrolled to detect a given and significant difference (and clinically meaningful). It is true that the number of cases ($n=24$) could have been higher, although the proof of a clinical relevance is given by the strong validation that we have performed in external datasets.

This does not exclude, however, the possibility that other “signature markers” might exist that are also significant in terms of clinical relevance. In our new experiments, and included in the revised manuscript, we report that: #1. a signature of 13 proteins (13P), representing an overlap between 33P and our validation data set when using

12 cell lines, was found to be even slightly stronger than the 33P in multivariate analysis, demonstrating that an expanded cell line panel strengthens the stratification impact. #2. in bioinformatical processing of the 33P signature further, by a stepwise leave-one-out reduction algorithm, we identified a subsignature of 18 proteins (within the 33P) that was stronger than the core signature 33P. These shorter subsignatures of 33P may be more easily applied and translated to clinical assays.

These findings, which have been included in the revised manuscript, further support the validity of our findings.

The new paragraph on the validation BCCL experiments (#1.) is presented under comment 1.1.

The new paragraph on bioinformatical processing (#2.) of 33P in the revised manuscript is as follow:

“Subsequently, we asked whether any of the 33 proteins were more important than others in terms of their impact on patient survival. Using the METABRIC-Discovery dataset (n=852), we applied a reduction algorithm, assuming that not all proteins in 33P would be equally strong. Thus, the 33P signature was reduced by recursively leaving one gene/protein out of the signature and then testing the predictive strength of the remaining N-1 genes/proteins in a survival analysis (Q1-3 vs. Q4). The strongest N-1 signature (lowest log-rank p-value) was retained, and the process was repeated until only one gene remained. The reduced version of 33P that showed the strongest effect on survival ($p=4.3\times 10^{-17}$, compared to baseline 33P $p=1.0\times 10^{-8}$) was these 18 proteins: CDC37, COL5A1, CTSB, GAPDH, GRB2, HNRNPA1, HNRNPD, HNRNPF, HSPA4, HSPA9, IDH1, LDHA, MYL6, P4HB, PGK1, RRBP1, SET, VASP (**Supplementary Figure 8**). These 18 proteins also showed a strong separation of the upper quartile patients (Q4) in the luminal A subgroup, and the prognostic impact was validated in KMplotter ($p<1.0\times 10^{-16}$; n=2032), also in the luminal A subgroup ($p=0.00015$; n=631).”

Reviewer #4, expertise in breast cancer subtypes/TME (Remarks to the Author):

The paper of X and Y entitled “Hypoxia induced responses are reflected in the stromal proteome and provide essential information for breast cancer stratification” studies the proteins that are secreted in cell lines of luminal and basal nature, (1877) proteins altogether. In the first part of the study the authors compare the protein lists in the two types of cell lines, each grown in normoxic and hypoxic conditions. This leads to a number of comparisons (two types of cell lines, grown in 2 different conditions), which reveal that different proteins are secreted in each 4 situations (luminal/basal like and normoxic/hypoxic). I like Supplementary figure 1 and propose to bring it forward in the paper as well as build the presentation of the data according to where on the workflow one is. GSEA was performed to identify the biological functions, and they were found strikingly different in luminal vs basal, i.e. hypoxia triggers different signaling and metabolic pathways in both subtypes of BC. Only 7 proteins overlapped and showed increased abundance in both subtypes after hypoxia.

Then the authors performed the same proteomic analysis on laser captured

microdissection specimens from breast cancer and identified 4,157 proteins in the epithelial tumor cell compartment, and 2,150 proteins in stromal samples. Of these 33 genes overlapped with the genes found from the cell lines experiment.

Then the authors “reverse-translate” these to mRNA and start interrogating larger mRNA compendiums (Metabric) using IPA to identify enrichment of genetic pathways and see if they can recapitulate the differential expression of these genes in the corresponding PAM50 subclasses and whether they bring difference in clinical presentation (survival).

The proteomics part of the paper (cell lines and microdissection) is rather straightforward and brings novel insight.

4.0. RESPONSE TO GENERAL COMMENTS:

We thank the reviewer for these comments. We agree that the workflow could be moved forward to clarify early on the structure of the project, and we have now done so in the revised version of the paper and the figure is now presented as **Figure 1**. We have also included the extraction of the 33P signature in the workflow to clarify our approach (in response to comment 1.6).

4.1. COMMENT:

I would suggest in this part the authors to separate better what is expected and confirmatory from what was known from before: The proteins that separate luminal from basal-line- are any of them they part of PAM50?

4.1. RESPONSE: also #4.5.

We thank the reviewer for this comment, and we agree that this is an important point. Notably, there were no overlapping (gene/proteins) between the PAM50 signature and the 33P signature of stromal hypoxia-related proteins generated in this work.

When we compare the proteins representing significant and differential secretion or expression between basal-like and luminal-like proteins, we find that the majority of these are not part of the PAM50 signature proteins, with only a few proteins overlapping (see **Breast cancer subtypes** in **Table 2** in this letter).

As predicted, some of the proteins are in common, and there are more overlapping proteins for the microdissected epithelium (6 proteins; FOXA1, ERBB2, MAPT, NAT1, PHGDH, KRT5) than for the microdissected stromal compartment (1 protein; PHGDH). This highlights that the PAM50 signature is (mainly) epithelial cell-based. Notably, one of our research questions has been whether the tumor stroma could provide additional information relevant for tumor classification and patient stratification, beyond what is already available.

The same was the case for the normoxic (baseline) and hypoxic secretomes in comparing the subtypes: PAM50 has more genes overlapping with the baseline (3 proteins; EGFR, CDH3, SLC39A6) than the hypoxic (1 protein; EGFR) cell secretomes, and hence may be lacking hypoxic information in the subtype stratification, as presented in **Table 2** in this letter.

An overview of overlapping proteins between significantly and differentially secreted or expressed proteins and known signatures reflecting breast cancer subtypes, hypoxia, and stromal features, are presented in **Table 2** in this letter to better separate the known information from our novel findings, as also commented on in the following comments (comment 4.1-4.5). In the revised manuscript, information on this and part of **Table 2** in this letter has now been included in a revised version as **Supplementary Table 10**.

Table 2: Proteins in common for differentially secreted or expressed proteins with signatures for breast cancer subtypes, hypoxia, and stromal features.

Breast cancer subtypes ⁽¹⁾		
	Overlapping genes/proteins	Signature/gene set
Cell secretome, baseline (464 proteins)	EGFR, CDH3, SLC39A6	PAM50 (50 genes)
Cell secretome, hypoxia (600 proteins)	EGFR	PAM50 (50 genes)
Microdissected tumor epithelium (844 proteins)	FOXA1, ERBB2, MAPT, NAT1, PHGDH, KRT5	PAM50 (50 genes)
Microdissected tumor stroma (473 proteins)	PHGDH	PAM50 (50 genes)
Oxygen conditions/hypoxia ⁽²⁾		
	Overlapping genes/proteins	Signature/gene set
Breast cancer hypoxia response proteins (150 proteins)	–	PAM50 (50 genes)
	GAPDH, AK2	Halle, 2012 (31 genes)
	VEGFA, ANGPTL4, LDHA, PGK1	Eustace, 2013 (26 genes)
	RNASE4	Ragnum, 2015 (32 genes)
	ANGPTL4, LDHA, S100A4, COL5A1, PRDX5, MYH9, GAPDH, VEGFA, GPI, FBP1, PGK1	Hallmark hypoxia (200 genes)
Stromal hypoxia ⁽³⁾		
	Overlapping genes/proteins	Signature/gene set
33P stromal-based hypoxia signature (33 proteins)	Breast cancer subtypes	
	–	PAM50 (50 genes)
	Hypoxia signatures	
	AK2, GAPDH	Halle, 2012 (31 genes)
	LDHA, PGK1	Eustace, 2013 (26 genes)
	–	Ragnum, 2015 (32 genes)
	LDHA, COL5A1, PGK1, S100A4, GAPDH	Hallmark hypoxia (200 genes)
	Proliferation signatures	
	GAPDH	OncotypeDx; Paik, 2004 (21 genes)
	–	PCNA proliferation signature; Venet, 2011 (131 genes)

Glycolysis	
COL5A1, MDH2, LDHA, IDH1, PGK1	Hallmark glycolysis (200 genes)
Vascular proliferation	
–	Hu, 2009 (13 genes)
–	Stefansson, 2015 (32 genes)
EMT and stemness	
–	Jechlinger, 2003 (128 genes)
P4HB, GAPDH, AK2	Pece, 2010 (299 genes)
–	Kruger, 2017 (44 genes)
CTSB	Luminal progenitor signature; Lim, 2009 (626 genes)
AK2, HNRNPA1	Mature luminal signature; Lim, 2009 (990 genes)

⁽¹⁾ Breast cancer subtypes: proteins significantly different between basal-like vs. luminal-like breast cancer cell lines or patients. Student's t-test, significance level $p < 0.05$.

⁽²⁾ Oxygen conditions/hypoxia: breast cancer hypoxia response proteins (150 proteins) consist of proteins with increased secretion in response to hypoxia; proteins with significantly higher secretion from hypoxic vs. normoxic breast cancer cell line secretomes. Student's t-test, significance level $p < 0.05$.

⁽³⁾ Stromal hypoxia: 33P stromal-based hypoxia signature (33 proteins) derived from breast cancer hypoxia response proteins and stromal proteome information.

4.2. COMMENT:

The proteins that separate the stromal from the epithelial: how many are known from before?

4.2. RESPONSE:

Different expression patterns between tumor stroma and tumor epithelium were expected, and in our analysis, we found 283 proteins that were differentially expressed between the basal-like and luminal-like breast cancer subtypes in the tumor stroma only (not in the tumor epithelium). Several of these are well known proteins such as MMP2, TIMP3, PDGFRB and FAP. When further exploring the cellular compartment of these proteins by gene ontology analysis, we found a significant overrepresentation of proteins in 'Extracellular matrix' (GOID 31012, $FDR = 6.60 \times 10^{-15}$) and 'Extracellular space' (GOID 5615, $FDR = 1.37 \times 10^{-57}$), as well as involvement in processes of 'Extracellular matrix organization' (GOID 30198, $FDR = 5.03 \times 10^{-5}$) and 'Collagen fibril organization' (GOID 30199, $FDR = 7.81 \times 10^{-4}$). This information has now been included as a new paragraph in the revised manuscript.

As microdissected stroma represents 'non-tumor areas', these regions are expected to also contain intracellular proteins, with a higher relative abundance of extracellular matrix proteins. Clearly, microdissection of tumor epithelial and stromal compartments allows for more precise separation than whole tissue investigations.

The new paragraph included in the manuscript text is as follow:

"When further exploring these proteins by gene ontology analysis, we found a significant overrepresentation of proteins in the cellular components 'Extracellular matrix' (GOID 31012, $FDR = 6.60 \times 10^{-15}$) and 'Extracellular space' (GOID 5615,

FDR=1.37×10⁻⁵⁷), as well as involvement in processes of ‘Extracellular matrix organization’ (GOID 30198, FDR=5.03×10⁻⁵) and ‘Collagen fibril organization’ (GOID 30199, FDR=7.81×10⁻⁴).”

4.3. COMMENT:

The proteins that separate normoxic from hypoxic: how many are known from before and part of known and widely signatures of hypoxia?

4.3. RESPONSE:

We would like to thank the reviewer for this comment and refer to the table presented under comment #4.1. (**Table 2** in this letter). As presented in this table, only a few proteins are overlapping between the 150 hypoxia-increased proteins from our secretome experiments and previously found hypoxia signatures, investigated in this work (2 proteins, Halle 2012; 4 proteins, Eustace 2013; 1 protein, Ragnum 2015). This point to the complexity and heterogeneity of the hypoxia response as mapped by signatures applied on clinical samples. **Table 2** in this letter is included in the revised manuscript as **Supplementary Table 10**.

Further, from our IPA analysis, we found 15 of the 150 hypoxia-upregulated proteins to be HIF1A targets. Although it is expected that at least some of the differentially secreted proteins are HIF targets, as indicated by the IPA analysis, there are not only HIF targets that are differentially secreted and take part in the hypoxia response. This has been added to the Results text in the revised manuscript.

We found HIF1A but not HIF2A in top ranked upstream transcription factors, supporting that the proteins identified in our experiments are among the initial hypoxia response proteins³⁸. This has been included in the revised manuscript.

In total, 11 of our 150 hypoxia-upregulated proteins overlapped with the ‘Hallmarks hypoxia’ gene set (MSigDB), representing well known and established proteins expressed under hypoxic conditions (overlap presented in **Table 2** in this letter under **comment 4.1**).

Taken together, these data indicate novel information on the early hypoxia response in breast cancer, that may not be solely reflected in HIFs or well-known markers. However, our findings are correlated to hypoxia signatures (see also comment 4.4) and showing patterns as expected under hypoxic conditions, by identified HIF targets and “hallmark hypoxia” proteins. We believe that our findings shed light on the breast cancer hypoxia response in luminal-like and basal-like cells and the message that is communicated to the surrounding microenvironment.

4.4. COMMENT:

When one performs correlation to hypoxia signatures, has one checked that the latter do not consist of the same genes?

4.4. RESPONSE:

Thank you for this important comment. We have correlated the 33P signature with three hypoxia related signatures from the literature: Halle (2012), Eustace (2013) and

Ragnum (2015) (referred to in the manuscript). The latter did not have any overlapping genes, but Halle (AK2, GAPDH) and Eustace (LDHA, PGK1) each had two overlapping genes. We have now performed new correlation analyses with both Halle and Eustace against the 33P signature, with exclusion of the overlapping genes. Both signature correlations are still highly significant ($p < 0.001$) and with similar Spearman's rank correlation coefficients (Halle: $\rho = 0.350$, previously $\rho = 0.405$; Eustace: $\rho = 0.591$, previously $\rho = 0.644$). We have included this information in the revised manuscript and kept the original plots with the complete 33P (**Supplementary Figure 6**).

The overlapping genes/proteins are presented in the table under comment 4.1. (**Table 2** in this letter, included as **Supplementary Table 10** in the revised manuscript).

4.5. COMMENT:

Of the genes comprising the PAM50 - how many proteins are in the epithelial and how many in the secreted compartment?

4.5. RESPONSE:

According to the plasma proteome database (PPD; <http://www.plasmaproteomedatabase.org/>)^{39, 40}, 38 of the 50 genes are reported in plasma or serum. Among the overlapping proteins with evidence of being detected in plasma or serum are EGFR (basal-like vs. luminal-like, baseline and hypoxic secretome) and CDH3 (basal-like vs. luminal-like, baseline secretome only). Further, two of these proteins (MMP11 and SFRP1) were detected in plasma by mass spectrometry analysis in the Human Protein Atlas – blood protein (Human Protein Atlas proteinatlas.org)⁴¹. However, among the 10 proteins not reported in serum or plasma, and hence not expected to find in the secretome, we find one protein (SLC39A6; basal-like vs. luminal-like, baseline secretome only). The overlapping proteins are presented in **Table 2** in this letter, and in connection to comment 4.1 regarding PAM50.

The manuscript has been updated to include this information in the baseline analysis of our secretome data.

4.6. COMMENTS:

My main criticism is to the last part of the paper, the validation in Metabric and related cohorts. While it is a valid argument and valuable finding that the 33 gene signature is associated to survival, altogether and in LumA, LumB and Basal separately as well as related clinic-histo-pathological parameters (tumor size, grade), one could do a lot more than apply IPA to identify signaling and metabolic circuits that encompass these 33 proteins (see methods like Bayesian network inference, SEEK, etc) as well as machine learning methods, which one can employ which other pathways and genes are generated using these 3 as a seed. This may answer questions like where these proteins are produced, what cell types etc.

4.6. RESPONSE:

We agree with the reviewer that exploration of molecular and clinico-pathological characteristics towards the identification of “signatures” is a relevant topic. Also, the application of artificial intelligence creates many unprecedented opportunities for

multidimensional data discovery and interrogation. Here, we used SEEK datasets to further validate our 33P signature score⁴².

The new paragraph on SEEK is as follows:

“We also applied SEEK to support our findings and found that 33P associated with triple-negative phenotype ($p=0.0006$) and high-grade breast cancer ($p<0.00001$) in two datasets (GSE45255.GPL96 and GSE4922.GPL96), as well as p53 mutations (GSE22093.GPL96; $p=0.038$); the same association was found in METABRIC-Discovery, also among luminal A cases ($p=0.02$). 33P was also higher in tumor tissue compared with normal tissue (GSE15852.GPL96) ($p=0.001$).”

To expand the functional characterization of 33P proteins, we performed a string-analysis (string-db.org), and found very strong connectivity between the proteins. In fact, 29 of the 33 proteins (88%) were included in one large network (included in the revised manuscript as **Supplementary Figure 9**). This is not unexpected since the 150 hypoxia proteins that 33P is derived from also showed high connectivity (83% in one large network, see **Figure 3** in revised manuscript). Interestingly, we found that 9 of the 33P proteins were associated with the “VEGFA-VEGFR2 signaling pathway” (WikiPathways, $p<0.001$). This supports our hypothesis that the 33P signature is involved in hypoxia, as VEGFA and hypoxia are closely linked. We have included this part in the revised manuscript.

Also, we used Cibersort (see response to comment #4.9) in an attempt to explore where these proteins are produced, including cell type information.

4.7. COMMENTS:

In fact at this stage of the analyses it is not clear why the authors want to restrict themselves to the 33 proteins, which represent the significant and unique differences between the subtypes in the stromal compartment? I.e. if one looks for validation in bulk mRNA expression data that comprise of all cells admixtures, why restrict oneself to the 33 proteins that are found in the stroma and one does not even know where they come from? Here one may go back and up the steps one came down to generate the 33 gene list and perform the same validation in Metabric (luminal vs basal, stromal vs epithelial, hypoxic vs normoxic).

4.7. RESPONSE:

We thank the reviewer for these important comments. We agree that there might be additional ways to interrogate these secretome and tissue data that we have obtained. Notably, we initially asked whether there are global stromal proteome profiles related to hypoxia that could represent additional and clinically useful information after validation; it is clear that many relevant questions would not be extensively studied by our focused approach and these would be suitable for follow-up studies, especially when we have now established an extended secretome data set including 12 breast cancer cell lines at normoxia and hypoxia (we predict that this proteome data set might prove useful also in future studies since there are not many similar resources available, to our knowledge, for direct comparison).

Basically, one interpretation of the reviewer's question is whether the 33P is the final (and only) signature of potential clinical value related to our aim and approach. We agree that it is quite possible to generate multiple signatures based on our setup. We have assumed though, within our approach, that a combined interrogation (as we did) would increase the signal-to-noise ratio.

Notably, the 33P signature was derived from the overlap of 150 hypoxia-upregulated proteins and the 283 stroma derived proteins (see workflow of signature extraction in **Figure 1** in revised manuscript). To examine the "uniqueness" of this 33P signature compared to a random selection of 33 proteins from a pool of the 150 and 283 proteins, we performed a random selection permutation analysis and found that 33P was significantly stronger than expected by random chance ($p < 0.0001$; **Supplementary Figure 3**).

Our extended experiments indicated that a shorter subset of the 33 proteins, *i.e.*, 13P, was slightly stronger as a prognostic factor, in particular among luminal A cases, and these data have been included in the revised manuscript.

Also, we asked whether any of the 33 proteins were more important than others in terms of their impact on patient survival. Using the METABRIC-Discovery dataset ($n=852$), we applied a reduction algorithm, assuming that not all proteins in 33P would be equally strong, and the 33P-signature was reduced by recursively leaving one gene/protein out of the signature, resulting in a shorter signature (of 18 proteins, which was also significant in KMplotter (as validation)). This information has been included in the revised manuscript in new paragraphs, presented under comment 1.1 (13P subsignature from validation experiments) and comment 3.6 (18 protein reduction algorithm).

Regarding hypoxia, direct data is not available for comparison, for example in the METABRIC cohorts. The purpose of our work was to introduce such "hypoxia response data" by combination of secretome information with our global stromal proteome data to avoid indirect modelling.

As to the comment on 33P that "*one does not even know where they come from?*", we agree that this is not necessarily known. We have assumed that many of these proteins are secreted from tumor cells, although some of them might also come from stromal cells and extracellular sources. We based our interpretation on the fact that these proteins can be found in the stroma and would indicate a potential functional role in this compartment. We think that the origin of the proteins is even more complicated for several other "signatures" in the literature derived from bulk tissues and which have been validated as useful and received regulatory approval for clinical management.

Regarding relations to specific cell types (cells of origin, and others), other techniques for spatial interrogation would be possible, like imaging mass cytometry, and suitable for follow-up studies.

Still, as commented on below (#4.9) we applied Cibersort and deconvolution in an attempt to illuminate the relation between 33P and certain cell types in the stroma compartment. This information has been included in the revised manuscript in a new paragraph (see comment 4.9).

4.8. COMMENTS:

Another critical moment is what the interaction to treatment response is and what part of the poor survival can be associated with poor treatment response. There are a number of datasets out there with known treatment, where one can look for interaction. produced?

4.8. RESPONSE:

Thank you for this very important comment. We agree that a potentially treatment-predictive value of 33P is very relevant from a clinical point of view, and we have now performed additional analyses to look into this, and revised the manuscript; this has now been added:

“To explore the potential interaction between 33P and various treatments, we applied the retrospective observational METABRIC-Discovery cohort (n=852) with information on endocrine treatment, chemotherapy, and radiation therapy. We initially performed stratified survival analyses (with/without treatment), and we found no difference for endocrine treatment or chemotherapy with respect to 33P, while different survival patterns were found for radiation therapy (yes vs. no) (**Figure 6a-b**). For those treated with radiotherapy, low values (lower quartile) of 33P were associated with significantly better survival than high values (upper quartile). Statistically, we found a significant interaction with radiotherapy for the prognostic value of 33P ($p=0.02$; HR=1.93 [1.21–3.30]). The diverging effect of radiotherapy was also significant in patients with luminal A breast cancer (**Figure 6c-d**). Potentially, the 33P signature may be applied to stratify patients for radiotherapy, with low 33P values predicting better radiation therapy response, although a prospective randomized clinical trial is needed to confirm this finding.”

4.9. COMMENT:

Perhaps with methods such as Cibersort or the like, one could see if this signature does not correlate to any cell type?

4.9. RESPONSE:

We appreciate this comment from the reviewer. Indeed, we used Cibersort to deconvolute bulk transcriptomic data from METABRIC, and we inferred the immune cell abundance for a subset of patients with basal-like cancers⁴³. In this exploratory subgroup analysis, we only included basal-like breast cancer samples since these are typically associated with higher levels of immune cell infiltration compared to the luminal subtypes, and the currently available Cibersort signature matrices are trained on triple-negative breast cancer. Based on the inferred immune cell abundance from Cibersort, patients were subsequently stratified using the 33P signature score (Q1-Q3 vs. Q4). We have added **Supplementary Figure 4**, showing lower number of B-cells and CD8+ cells in the worse outcome (Q4) group. This is consistent with published data showing a positive correlation between TILs and survival⁴⁴. Interestingly, we also found a fewer resting mast cells, and an increase in activated mast cells in the Q4-group. These data suggest an association between mast cell activation and poor outcome for breast cancer patients. We believe that these particular findings, although very preliminary, must be explored in subsequent studies.

The manuscript has been amended to reflect these changes and now describe the additional analyses and results. The following has been included in the revised manuscript:

“To illuminate potential associations between 33P and specific cell types in the tumor stroma, we used Cibersort⁴³ to deconvolute bulk transcriptomic data from METABRIC-Discovery (n=852). We inferred the immune cell abundance for a subset of patients with basal-like and triple-negative breast cancer, as the Cibersort signature matrices currently available⁴⁵ are trained on triple-negative, and not luminal-like, breast cancer. Thus, the basal-like patients were subsequently stratified using the 33P signature score (Q1-Q3 vs. Q4), and we observed lower number of B-cells and CD8-cells in the worse outcome (Q4) subgroup of 33P, indicating potential immune suppression (**Supplementary Figure 4**). Notably, we also found fewer resting mast cells, and an increase in activated mast cells associated with higher 33P. Our findings indicate an association between 33P and immune cell levels within the basal-like subtype.”

4.10. COMMENT:

In conclusion, I like the clear thought line in this paper and the analytic workflow, but I think that in the synthetic part, when thinking of what the results mean, the authors can use more of the data they have generated and not only the 33 gene list. The advantage of such a defined and limited number of genes is in the possibility to assess the state of hypoxia non-invasively. Is there any evidence that any of these proteins are also detectible in serum or plasma?

4.10. RESPONSE:

We thank the reviewer for this important comment. As this is considered one of the main strengths of secretome studies, we have investigated the 33 signature proteins in the plasma proteome database (PPD; <http://www.plasmaproteomedatabase.org/>)^{39, 40}. We found 32 of the 33 signature proteins are detected in serum or plasma in this database. The protein that was not found in PPD was Isoform 1 of Coatamer subunit epsilon (COPE). This protein is involved in endoplasmic reticulum to Golgi vesicle-mediated transport, which may be a possible explanation for our detection of this protein in our secretome data.

Additionally, the 33P signature proteins were looked at in the Human Protein Atlas – blood protein (Human Protein Atlas proteinatlas.org)⁴¹. Here, we found all signature proteins to be detected in plasma by mass spectrometry analysis. This information is now included in the revised manuscript.

The new paragraph is as follows:

“As one of the main strengths of secretome studies is the potential presence of such proteins in serum or plasma, we investigated the 33 proteins in the PPD^{39, 40} and found 32 of the 33 signature proteins (not in PPD: COPE). We further explored the Human Protein Atlas – blood protein⁴¹ and found all signature proteins to be detected in plasma by MS analysis.”

Unfortunately, we are not aware of suitable BC cohorts with information on these plasma proteins that could be applied. This would represent a very attractive follow-up study.

As we describe in the revised manuscript, in our extended cell experiments, we have identified a shorter version of the 33P that might potentially be attractive for development of a clinical assay (13P).

As a key finding in terms of potential clinical importance, our additional data on interactions with treatment have indicated a possibility that 33P could be applied as a predictive marker for radiation therapy, if sufficiently validated in the context of RCTs.

REFERENCES

1. Neve RM, *et al.* A collection of breast cancer cell lines for the study of functionally distinct cancer subtypes. *Cancer Cell* **10**, 515-527 (2006).
2. Dai X, Cheng H, Bai Z, Li J. Breast Cancer Cell Line Classification and Its Relevance with Breast Tumor Subtyping. *J Cancer* **8**, 3131-3141 (2017).
3. Kao J, *et al.* Molecular profiling of breast cancer cell lines defines relevant tumor models and provides a resource for cancer gene discovery. *PLoS One* **4**, e6146 (2009).
4. Holliday DL, Speirs V. Choosing the right cell line for breast cancer research. *Breast Cancer Res* **13**, 215 (2011).
5. Lehmann BD, *et al.* Identification of human triple-negative breast cancer subtypes and preclinical models for selection of targeted therapies. *J Clin Invest* **121**, 2750-2767 (2011).
6. Prat A, *et al.* Characterization of cell lines derived from breast cancers and normal mammary tissues for the study of the intrinsic molecular subtypes. *Breast Cancer Res Treat* **142**, 237-255 (2013).
7. Nusinow DP, *et al.* Quantitative Proteomics of the Cancer Cell Line Encyclopedia. *Cell* **180**, 387-402 e316 (2020).
8. Ghandi M, *et al.* Next-generation characterization of the Cancer Cell Line Encyclopedia. *Nature* **569**, 503-508 (2019).
9. Perou CM, *et al.* Molecular portraits of human breast tumours. *Nature* **406**, 747-752 (2000).
10. Sorlie T, *et al.* Gene expression patterns of breast carcinomas distinguish tumor subclasses with clinical implications. *Proc Natl Acad Sci U S A* **98**, 10869-10874 (2001).

11. Semenza GL. Cancer-stromal cell interactions mediated by hypoxia-inducible factors promote angiogenesis, lymphangiogenesis, and metastasis. *Oncogene* **32**, 4057-4063 (2013).
12. Cox J, Mann M. MaxQuant enables high peptide identification rates, individualized p.p.b.-range mass accuracies and proteome-wide protein quantification. *Nat Biotechnol* **26**, 1367-1372 (2008).
13. Subramanian A, *et al.* A Next Generation Connectivity Map: L1000 Platform and the First 1,000,000 Profiles. *Cell* **171**, 1437-1452 e1417 (2017).
14. Zhang Q, Tang X, Lu QY, Zhang ZF, Brown J, Le AD. Resveratrol inhibits hypoxia-induced accumulation of hypoxia-inducible factor-1alpha and VEGF expression in human tongue squamous cell carcinoma and hepatoma cells. *Mol Cancer Ther* **4**, 1465-1474 (2005).
15. Perl A. mTOR activation is a biomarker and a central pathway to autoimmune disorders, cancer, obesity, and aging. *Ann N Y Acad Sci* **1346**, 33-44 (2015).
16. Salehi B, *et al.* The Therapeutic Potential of Apigenin. *Int J Mol Sci* **20**, (2019).
17. Rojo de la Vega M, Chapman E, Zhang DD. NRF2 and the Hallmarks of Cancer. *Cancer Cell* **34**, 21-43 (2018).
18. Sajadimajd S, Khazaei M. Oxidative Stress and Cancer: The Role of Nrf2. *Curr Cancer Drug Targets* **18**, 538-557 (2018).
19. Semenza GL, Nejfelt MK, Chi SM, Antonarakis SE. Hypoxia-inducible nuclear factors bind to an enhancer element located 3' to the human erythropoietin gene. *Proc Natl Acad Sci U S A* **88**, 5680-5684 (1991).
20. Wang GL, Jiang BH, Rue EA, Semenza GL. Hypoxia-inducible factor 1 is a basic-helix-loop-helix-PAS heterodimer regulated by cellular O₂ tension. *Proc Natl Acad Sci U S A* **92**, 5510-5514 (1995).
21. Maxwell PH, *et al.* The tumour suppressor protein VHL targets hypoxia-inducible factors for oxygen-dependent proteolysis. *Nature* **399**, 271-275 (1999).
22. Ivan M, *et al.* HIFalpha targeted for VHL-mediated destruction by proline hydroxylation: implications for O₂ sensing. *Science* **292**, 464-468 (2001).
23. Jaakkola P, *et al.* Targeting of HIF-alpha to the von Hippel-Lindau ubiquitylation complex by O₂-regulated prolyl hydroxylation. *Science* **292**, 468-472 (2001).
24. Schito L, Semenza GL. Hypoxia-Inducible Factors: Master Regulators of Cancer Progression. *Trends Cancer* **2**, 758-770 (2016).

25. Wicks EE, Semenza GL. Hypoxia-inducible factors: cancer progression and clinical translation. *J Clin Invest* **132**, (2022).
26. Vinaiphath A, Low JK, Yeoh KW, Chng WJ, Sze SK. Application of Advanced Mass Spectrometry-Based Proteomics to Study Hypoxia Driven Cancer Progression. *Front Oncol* **11**, 559822 (2021).
27. Cox TR, *et al.* The hypoxic cancer secretome induces pre-metastatic bone lesions through lysyl oxidase. *Nature* **522**, 106-110 (2015).
28. Yoon JH, *et al.* Proteomic analysis of hypoxia-induced U373MG glioma secretome reveals novel hypoxia-dependent migration factors. *Proteomics* **14**, 1494-1502 (2014).
29. Maia J, Caja S, Strano Moraes MC, Couto N, Costa-Silva B. Exosome-Based Cell-Cell Communication in the Tumor Microenvironment. *Front Cell Dev Biol* **6**, 18 (2018).
30. Rankin EB, Giaccia AJ. Hypoxic control of metastasis. *Science* **352**, 175-180 (2016).
31. Fruman DA, Chiu H, Hopkins BD, Bagrodia S, Cantley LC, Abraham RT. The PI3K Pathway in Human Disease. *Cell* **170**, 605-635 (2017).
32. Gonzalez-Angulo AM, *et al.* Open-label randomized clinical trial of standard neoadjuvant chemotherapy with paclitaxel followed by FEC versus the combination of paclitaxel and everolimus followed by FEC in women with triple receptor-negative breast cancer. *Ann Oncol* **25**, 1122-1127 (2014).
33. Shamis SAK, McMillan DC, Edwards J. The relationship between hypoxia-inducible factor 1alpha (HIF-1alpha) and patient survival in breast cancer: Systematic review and meta-analysis. *Crit Rev Oncol Hematol* **159**, 103231 (2021).
34. Le QT, Courter D. Clinical biomarkers for hypoxia targeting. *Cancer Metastasis Rev* **27**, 351-362 (2008).
35. Conklin MW, Keely PJ. Why the stroma matters in breast cancer: insights into breast cancer patient outcomes through the examination of stromal biomarkers. *Cell Adh Migr* **6**, 249-260 (2012).
36. Aggarwal N, Sloane BF. Cathepsin B: multiple roles in cancer. *Proteomics Clin Appl* **8**, 427-437 (2014).
37. Askeland C, *et al.* Stathmin expression associates with vascular and immune responses in aggressive breast cancer subgroups. *Sci Rep* **10**, 2914 (2020).
38. Liu Q, Palmgren VAC, Danen EH, Le Devedec SE. Acute vs. chronic vs. intermittent hypoxia in breast Cancer: a review on its application in in vitro research. *Mol Biol Rep* **49**, 10961-10973 (2022).

39. Muthusamy B, *et al.* Plasma Proteome Database as a resource for proteomics research. *Proteomics* **5**, 3531-3536 (2005).
40. Nanjappa V, *et al.* Plasma Proteome Database as a resource for proteomics research: 2014 update. *Nucleic Acids Res* **42**, D959-965 (2014).
41. Uhlen M, *et al.* The human secretome. *Sci Signal* **12**, (2019).
42. Zhu Q, *et al.* Targeted exploration and analysis of large cross-platform human transcriptomic compendia. *Nat Methods* **12**, 211-214, 213 p following 214 (2015).
43. Newman AM, *et al.* Determining cell type abundance and expression from bulk tissues with digital cytometry. *Nat Biotechnol* **37**, 773-782 (2019).
44. Mao Y, Qu Q, Chen X, Huang O, Wu J, Shen K. The Prognostic Value of Tumor-Infiltrating Lymphocytes in Breast Cancer: A Systematic Review and Meta-Analysis. *PLoS One* **11**, e0152500 (2016).
45. Craven KE, Gokmen-Polar Y, Badve SS. CIBERSORT analysis of TCGA and METABRIC identifies subgroups with better outcomes in triple negative breast cancer. *Sci Rep* **11**, 4691 (2021).

Reviewers' Comments:

Reviewer #1:

Remarks to the Author:

In the revised manuscript the authors added several cell lines to validate their analysis of four cell line secretomes. They also addressed some of my other comments. However, I still do not think the scope of the manuscript and the novelty are large enough to merit publication in Nature Communications. Breast cancer subtype differences have been studied extensively at the proteomic level in cell lines and tissues. Their analyses of secretomes and hypoxia do not seem to add any substantial novelty in terms of the functional and metabolic differences between subtypes. Furthermore, I do not support the emphasis on the clinical potential of the 33P signatures. Translation of proteomics data to the clinic would require analysis of hundreds-thousands of clinical samples, and will surely not be based primarily on a couple of cell lines. Altogether, I do not think the novelty and scope of the manuscript are sufficient for publication in Nature Communications.

Reviewer #3:

Remarks to the Author:

The authors have made a good faith attempt to respond to previous reviews, and have successfully addressed most of the technical, experimental design, rigor and reproducibility issues raised. However, the main criticism, that the results do not significantly add to the large body of published data on the contribution of hypoxia and tumor microenvironment to breast cancer malignancy, has not been addressed satisfactorily. Importantly, the results remain correlative without verification of key results. For example, the radiation resistance effects and purported central role of NFR2 are interesting, but there is no experimental verification of these results. The main contribution is identifying a proteomic signature that may identify a subset of recurrent patients, especially the luminal B subclass. It is difficult to see how the sophisticated proteomic approaches used to identify this signature can be easily moved to clinical practice.

In conclusion, the work is correlative and does not add mechanistic details that significantly increase knowledge on hypoxia and/or tumor microenvironment in breast cancer, and this does not meet the rigorous criteria needed for publication in this journal.

Reviewer #4:

Remarks to the Author:

I have now read through the comments to all 4 reviewers and the paper with new figures and figure legends. I am satisfied with the answers to my comments and see how they corroborate the comments of others. I am happy to see that my suggestions have brought new insights in known signatures (f.ex the PAM50 presence in the secretome, but lack of ability to reflect the effect of induced hypoxia) and less known- the potential role of the mast cells in prognosis (which we also see) and radiation. The shrinking of the 33signature to 13 is also valuable.

Now following thorough all reviewers comments and adding all required information the authors have as it often happens a new challenge- how to say the same in double so little space. It is in my view possible to shorten the text considerably, which will only improve its quality and clarity.

Reviewer #5:

Remarks to the Author:

This manuscripts represents a meaningful addition to literature, but needs attention to a few conceptual issues.

The novelty is mainly related (i) understanding of hypoxia secretome, (ii) prognostic value of stromal hypoxia signature, (iii) predictive value of stromal hypoxia signature for radiotherapy. Nothing that was found in this study was surprising, nevertheless it represents a significant resource for investigators in the field and may stimulate further research.

The issues are as follows:

1) The study design is conceptually unclear to me. Why are we looking at the overlap between cancer cell "secretome" and stromal cell proteome? If the question is whether stromal hypoxia is prognostic, why was not stromal proteome under hypoxia studied?

2) Among the "secreted" proteins discovered are numerous non-secreted proteins. There was particular enrichment of metabolic proteins, as highlighted throughout the manuscript. One might assume that the presence of these intracellular proteins in the media reflects cell lysis. Their presence reflects not only hypoxia-induction, but also tolerance to hypoxia. If more cells die, you will have more intracellular proteins in the media. As such, the focus and novelty of this manuscript pertaining to "secretome" can be debated. This limitation should be reflected in data interpretation, clearly acknowledged, and discussed.

3) The prognostic value in radiotherapy cohorts is interesting but needs further analysis. First, there is no information of the clinical cohorts that needs to accompany all these patient data. Are there clinical differences between patients treated with RT versus others that could underlie the unique prognostic value of P33? This seems likely. Second, although hypoxia is well known to mitigate local control after radiotherapy due to the radioresistance of hypoxic cells, breast cancer is a unique situation since radiotherapy most commonly is given post-surgery to address occult microscopic disease. I assume this is the case for this cohort? If so, it is unclear that hypoxia in the stroma of a (pre-)surgery specimen would impact the RT response of microscopic unresected disease post-surgery. Rather, perhaps, presence of hypoxia has stimulated metastasis (out-of-RT-field) and patients fail due to distant metastasis? Authors should analyze differences between patient cohorts, clarify treatment schedules and reasons for failure if possible (data are available), or at minimum enhance the discussion of significance of this finding.

Minor: KMs should indicate in legend what colors reflect P33 high/low

RE: NCOMMS-21-47819A

Hypoxia induced responses are reflected in the stromal proteome and provide essential information for breast cancer stratification

RESPONSE TO INDIVIDUAL REVIEWERS' COMMENTS

Reviewer #1

1.1. COMMENTS:

In the revised manuscript the authors added several cell lines to validate their analysis of four cell line secretomes. They also addressed some of my other comments. However, I still do not think the scope of the manuscript and the novelty are large enough to merit publication in Nature Communications.

Breast cancer subtype differences have been studied extensively at the proteomic level in cell lines and tissues. Their analyses of secretomes and hypoxia do not seem to add any substantial novelty in terms of the functional and metabolic differences between subtypes.

Furthermore, I do not support the emphasis on the clinical potential of the 33P signatures. Translation of proteomics data to the clinic would require analysis of hundreds-thousands of clinical samples, and will surely not be based primarily on a couple of cell lines.

1.1. RESPONSE:

We thank the reviewer for these comments. In our opinion, the data in our manuscript represent an addition to what we know about the global proteome in breast cancer microenvironment and in particular with a link to hypoxia, including subtype differences. The “overlap” approach between cell line data and clinical samples has, to our knowledge, not been previously performed. ***We also refer to our arguments in the previous response letter (R1).***

Regarding the “clinical potential”, this is still an open question. We believe that our findings would make sense in the context of radiation and deserve follow-up studies. In particular, such data from retrospective observational cohorts would need to be confirmed in randomized clinical trials.

We have now, in the manuscript, toned down our conclusions regarding the clinical significance of our proteomic signature, and we have included a paragraph in the manuscript where we discuss the limitations of our experimental design and data interpretation.

Reviewer #3

3.1. COMMENTS:

The authors have made a good faith attempt to respond to previous reviews, and have successfully addressed most of the technical, experimental design, rigor and reproducibility issues raised. However, the main criticism, that the results do not significantly add to the large body of published data on the contribution of hypoxia and tumor microenvironment to breast cancer malignancy, has not been addressed satisfactorily.

Importantly, the results remain correlative without verification of key results. For example, the radiation resistance effects and purported central role of NRF2 are interesting, but there is no experimental verification of these results.

3.1. RESPONSE:

We thank the reviewer for these comments. For the first comment, we refer to our response above (**response 1.1**).

For the second part of the comment, we present an integrated approach with combined data from cell line hypoxia experiments and clinical samples, and we further present validation data from retrospective patient cohorts.

In the manuscript, we have now discussed the limitations of our experimental design and data interpretation, and we have toned down our conclusions regarding the clinical significance of our proteomic signature.

Regarding NRF2, we already included a discussion of our findings; we agree that more studies and a more mechanistic approach would be needed to fully understand the role of NRF2. This is now added in the Discussion.

3.2. COMMENTS:

The main contribution is identifying a proteomic signature that may identify a subset of recurrent patients, especially the luminal B subclass. It is difficult to see how the sophisticated proteomic approaches used to identify this signature can be easily moved to clinical practice.

3.2. RESPONSE:

We thank the reviewer for these comments. As we have already mentioned in the manuscript, the shorter version of 33P (13P) might be a candidate signature for translation, if verified in further studies. Although discovered in our proteomic setup, our data already indicate that 13P could be validated (in METABRIC) based on *mRNA values from whole tissue samples* (as mentioned in our manuscript). It might then be feasible to establish an assay for practical use (as has been done with other mRNA-based signatures).

Reviewer #4

4.1. COMMENTS:

I have now read through the comments to all 4 reviewers and the paper with new figures and figure legends. I am satisfied with the answers to my comments and see how they corroborate the comments of others. I am happy to see that my suggestions have brought new insights in known signatures (f.ex the PAM50 presence in the secretome, but lack of ability to reflect the effect of induced hypoxia) and less known- the potential role of the mast cells in prognosis (which we also see) and radiation. The shrinking of the 33signature to 13 is also valuable.

4.1. RESPONSE:

We thank the reviewer for these comments.

4.2. COMMENTS:

Now following thorough all reviewers' comments and adding all required information the authors have as it often happens a new challenge- how to say the same in double so little space. It is in my view possible to shorten the text considerably, which will only improve its quality and clarity.

4.2. RESPONSE:

We thank the reviewer for these comments. We have made efforts to shorten the text somewhat, to improve quality and clarity. At the same time, we have added some text in the Discussion on limitations and possibilities of our approach (see also response 1.1. above), and relevant methodological limitations (see also response 5.2. below). (**see also response 1.1. above**).

Reviewer #5

5.1. COMMENTS:

This manuscripts represents a meaningful addition to literature, but needs attention to a few conceptual issues.

The novelty is mainly related (i) understanding of hypoxia secretome, (ii) prognostic value of stromal hypoxia signature, (iii) predictive value of stromal hypoxia signature for radiotherapy. Nothing that was found in this study was surprising, nevertheless it represents a significant resource for investigators in the field and may stimulate further research.

The study design is conceptually unclear to me. Why are we looking at the overlap between cancer cell "secretome" and stromal cell proteome? If the question is whether stromal hypoxia is prognostic, why was not stromal proteome under hypoxia studied?

5.1. RESPONSE:

We thank the reviewer for these comments on the value of our study. Our focus has been on the tumor microenvironment in breast cancer, tumor subtype differences, and effects of hypoxia. Regarding the overlap approach, we asked whether cell line secretome findings (after hypoxia) could also be observed in real tumor samples, assuming that proteins present in the tumor stroma (tumor microenvironment) might be of particular importance. We were able to show that such proteins (the 33P signature and the shorter 13P version) could be validated for their "clinical relevance"

in the external METABRIC and KMplotter cohorts (even based on mRNA values in whole tissue samples).

Apart from what we have already presented, we do not fully understand the last part of the comment – on “*why was not stromal proteome under hypoxia studied*”. According to our approach, we used cell lines under hypoxia to interrogate hypoxia patterns in the stroma, rather than using other hypoxia surrogate markers (e.g., by IHC) to stratify the clinical samples before proteomics; this would be another possible strategy although we assumed with lower sensitivity. Stratification by such markers would be indirect and might reflect other responses than just hypoxia.

5.2. COMMENTS:

Among the “secreted” proteins discovered are numerous non-secreted proteins. There was particular enrichment of metabolic proteins, as highlighted throughout the manuscript. One might assume that the presence of these intracellular proteins in the media reflects cell lysis. Their presence reflects not only hypoxia-induction, but also tolerance to hypoxia. If more cells die, you will have more intracellular proteins in the media. As such, the focus and novelty of this manuscript pertaining to “secretome” can be debated. This limitation should be reflected in data interpretation, clearly acknowledged, and discussed.

5.2. RESPONSE:

We thank the reviewer for these comments. We have now provided additional information to address the issues of *cell lysis* and *protein “location”*, and we propose some additional text in the Results and Discussion parts.

First, we would like to point out that the cancer secretome is defined as all proteins or factors that are secreted or released from the cancer cells, including extracellular vesicles, and that the modes of secretion in cancer might also be different from non-cancerous cells.

In response to the reviewer’s comments on cell lysis, we have included data on viability of cells conditioned under normoxic and hypoxic conditions. We find high and similar viability of cells, with no significant difference in viability between cells conditioned at normoxia and hypoxia, in either the luminal-like or basal-like cell lines (Mann-Whitney U test, significance level $p < 0.05$) (**figure below**).

Many intracellular proteins also have functions outside the cell, and we therefore looked at the number of proteins belonging to the extracellular region. Gene ontology showed significant enrichment of proteins in the extracellular region compared to random (GO:0005576; all proteins identified in discovery BCCLs (our first 4 cell lines), $FDR = 1.16 \times 10^{-229}$; all proteins identified in validation BCCLs (our panel of all 12 cell lines), $FDR = 2.75 \times 10^{-70}$).

The following text has been included in the *Results* section of the revised manuscript:

As we observed several intracellular proteins in our secretomes, we investigated cell viability and found this to be high, with no significant difference between cells conditioned at hypoxia and normoxia (average viability at hypoxia: 92.2%; normoxia: 93.9%; $p=ns$), in either the luminal-like (hypoxia: 95.9%; normoxia: 96.5%; $p=ns$) or basal-like cell lines (hypoxia: 88.5%; normoxia 91.4%; $p=ns$; Mann-Whitney U test). Further, gene ontology analysis of our secretome proteins showed significant enrichment of proteins in the extracellular region compared to random (GO:0005576; all proteins, discovery BCCLs , $FDR=1.16 \times 10^{-229}$).

The following text has been included in the *Discussion* section of the revised manuscript:

In our cell line experiments, we observed several of the secretome proteins that are normally found in the intracellular compartment. Although this could potentially reflect cell death, we found high viability with no significant viability difference between hypoxic and normoxic conditions when stratified by luminal-like and basal-like subtypes. Notably, intracellular cytosolic proteins may be contained in extracellular vesicles, and intracellular proteins may have extracellular functions. Also, secretion modes may be different in cancer cells compared to non-cancerous cells, and this could possibly explain why we detect intracellular proteins in our secretome data.

5.3. COMMENTS:

The prognostic value in radiotherapy cohorts is interesting but needs further analysis.

First, there is no information of the clinical cohorts that needs to accompany all these patient data. Are there clinical differences between patients treated with RT versus others that could underlie the unique prognostic value of P33? This seems likely.

Second, although hypoxia is well known to mitigate local control after radiotherapy due to the radioresistance of hypoxic cells, breast cancer is a unique situation since radiotherapy most commonly is given post-surgery to address occult microscopic disease. I assume this is the case for this cohort? If so, it is unclear that hypoxia in the stroma of a (pre-)surgery specimen would impact the RT response of microscopic unresected disease post-surgery.

Rather, perhaps, presence of hypoxia has stimulated metastasis (out-of-RT-field) and patients fail due to distant metastasis? Authors should analyze differences between patient cohorts, clarify treatment schedules and reasons for failure if possible (data are available), or at minimum enhance the discussion of significance of this finding.

5.3. RESPONSE:

We thank the reviewer for these important comments.

First, we agree with the reviewer that the RT+ and RT- groups are expected to be different, as is often the case in retrospective observational studies, and such differences are then linked to whether RT is given or not. We here show (below) how these subgroup are different in the METABRIC-Discovery cohort which we used to explore the possibilities of treatment interactions. To some extent, such differences can be accounted for by doing multivariate survival analyses, where (in this case) the interaction term between 33P and RT was still significant when added in addition “background” variables like tumor diameter, histologic grade and lymph node status (in Cox’ analysis). Still, as we previously pointed out in the manuscript, ***the possibility of 33P being a predictive marker for response to radiation treatment would need verification in a randomized clinical trial setting.***

Second, regarding the issue of “post-surgery RT”, we have assumed that even microscopic cancer remnants would have phenotypic characteristics which are similar to (or reflecting) the features of the main tumor bulk (already removed by surgery). Our data would be consistent with such an assumption.

Third, we also agree with the reviewer that this might be a possibility. Regarding treatment schedules, we have not been able to find information on the RT schedule in METABRIC-Discovery cohort.

5.4. COMMENTS:

Minor: KMs should indicate in legend what colors reflect P33 high/low.

5.4. RESPONSE:

We thank the reviewer for this comment. The figure legends have been updated to include information on which groups or quartiles that are 33P-high (Q4) and 33P-low (Q1-3).